# Learning Grouped Lattice Vector Quantizers for Low-Bit LLM Compression

**Xi Zhang**[1]     **Xiaolin Wu**[3]     **Jiamang Wang**[2]     **Weisi Lin**[1]✉

[1]Nanyang Technological University    [2]Alibaba Group    [3]Southwest Jiaotong University

{xi.zhang, wslin}@ntu.edu.sg

## Abstract

Large Language Models (LLMs) have demonstrated remarkable capabilities but typically require extensive computational resources and memory for inference. Post-training quantization (PTQ) can effectively reduce these demands by storing weights in lower bit-width formats. However, standard uniform quantization often leads to notable performance degradation, particularly in low-bit scenarios. In this work, we introduce a *Grouped Lattice Vector Quantization (GLVQ)* framework that assigns each group of weights a customized lattice codebook, defined by a learnable generation matrix. To address the non-differentiability of the quantization process, we adopt Babai rounding to approximate nearest-lattice-point search during training, which enables stable optimization of the generation matrices. Once trained, decoding reduces to a simple matrix-vector multiplication, yielding an efficient and practical quantization pipeline. Experiments on multiple benchmarks show that our approach achieves a better trade-off between model size and accuracy compared to existing post-training quantization baselines, highlighting its effectiveness in deploying large models under stringent resource constraints. Our source code is available on GitHub repository: https://github.com/xzhang9308/GLVQ.

## 1 Introduction

Large language models (LLMs) have achieved extraordinary success across a wide range of natural language processing tasks, including machine translation, question answering, text generation, and summarization [55, 12, 43]. These models, exemplified by the GPT [43], BERT [12], Llama [50], Qwen [2] and DeepSeek [32], are at the forefront of AI innovation. Beyond supervised pretraining, reinforcement learning techniques, most notably reinforcement learning from human feedback (RLHF), have further enhanced LLMs by aligning them with human preferences and improving their safety, helpfulness, and controllability [49, 40, 3, 33, 34, 16]. However, their exceptional performance comes at exorbitant computation and memory costs, which hinders their deployment on edge devices and in other resource-limited scenarios [7]. For instance, state-of-the-art models with billions of parameters often require hundreds of gigabytes of memory and expensive hardware for inference [5, 43], posing serious challenges for scalability and accessibility.

Quantization has emerged as a key technique to alleviate the resource burdens by reducing the precision of model weights from 16-bit or 32-bit floating-point representations to low-bit integers [28, 35]. This significantly reduces the memory footprint and computation load, improving model economy and inference speed. A relatively simple method is Post-training quantization (PTQ), which quantizes a pre-trained LLM model without retraining [29, 18]. However, PTQ has difficulties in achieving ultra-low bit-widths, such as 2 or 3 bits per weight [9]. Severe quantization errors can lead to significant degradation in model performance, such as higher perplexity in language tasks [18] or noticeable drops in downstream quality, thereby limiting the practical utility of low-bit quantization.

---

✉ Corresponding author.

39th Conference on Neural Information Processing Systems (NeurIPS 2025).

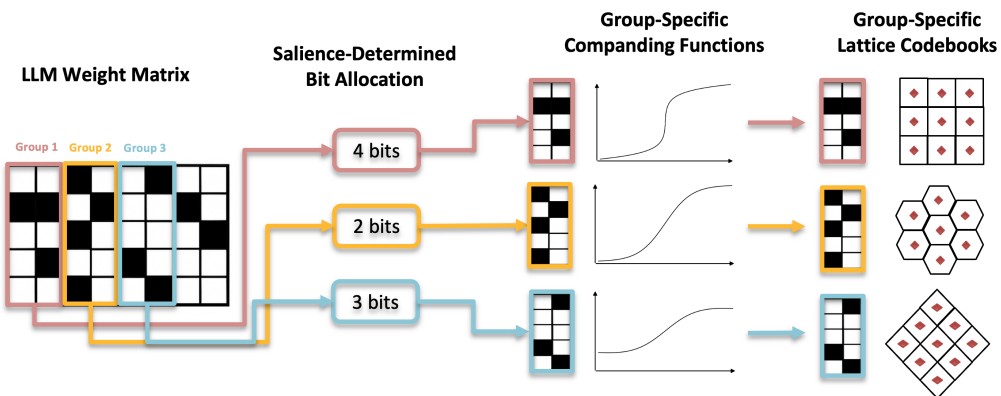

Figure 1: An overview of the GLVQ framework. The LLM weight matrix is partitioned into column-wise groups, each assigned a salience-determined bitwidth. Each group undergoes a tailored companding transformation and is quantized using a learned group-specific lattice codebook, enabling fine-grained control over precision and compression efficiency.

Recently, some progresses have been made in LLM weight quantization, including vector quantization (VQ) techniques [51, 14, 52] and salience-aware bit allocation [23]. Vector quantization methods, such as QuIP# [51], leverage structured codebooks (e.g., E8 lattice) to improve quantization fidelity by aligning the codebook structure with the statistical properties of model weights. However, QuIP# is constrained by the use of fixed lattice designs across the entire model, which fails to account for the diversity of parameter distributions across different groups or layers. This uniform approach can lead to suboptimal quantization for certain groups, ultimately limiting both compression performance and overall model quality. In contrast, AQLM [14] proposes learning free-form VQs for different groups, which allows for more flexible quantization. However, this approach has the drawback that decoding requires the lookup operation, which is computationally more expensive than QUIP# and other existing PTQ methods.

In this paper, we propose a novel framework, *Grouped Lattice Vector Quantization (GLVQ)* (illustrated in Fig. 1), aiming to increase the LLM compression ratio by reducing quantization resolution to be below 3 bits per weight. Specifically, GLVQ end-to-end optimizes the geometric structure of the lattice codebook for each quantization group based on that group's unique parameter distribution. By shaping the lattice to fit the data distribution of each group, we reduce quantization distortion, preserve critical information better, and improve model fidelity over nonadaptive lattices under extreme compression regimes. Compared to free-form VQ used in AQLM [14], GLVQ employs a structured codebook, hence it enjoys the operational advantage of much lower quantization complexity. Further investigations reveal that LLM parameter distributions can be highly skewed or heavy-tailed. If this unfortunately happens, optimizing GLVQ alone has limited effect, because the lattice cells of each group have congruent shape, albeit optimized, and have regular configurations. To counteract this, we incorporate group-specific companding transformations. These functions reshape each group's parameter distribution into a more uniform space before applying the lattice quantizer, thereby reducing distortion in the low-magnitude regions and enabling the lattice to allocate codepoints more effectively. Together, group-specific lattice codebook optimization and companding form a powerful synergy that maintains high accuracy while significantly reducing bit-width requirements.

We validate our approach on popular LLM benchmarks, demonstrating its effectiveness in improving perplexity and inference efficiency compared to state-of-the-art quantization methods [18, 51, 14, 13]. Our results show that GLVQ not only enables higher compression ratios but also maintains robust performance across diverse language tasks, making it a practical solution for deploying LLMs in resource-constrained environments. Furthermore, our framework provides insights into the interplay between parameter distribution and quantization fidelity, opening new directions for research in efficient model compression.

This paper makes the following key contributions:

- We propose a novel grouped lattice vector quantization approach that dynamically adapts lattice structures to fit the unique distribution of each weight group.

- We introduce a group-specific companding mechanism that reshapes each parameter distribution prior to lattice quantization, further reducing quantization distortion and improving overall performance.
- Extensive experiments on post-training quantization benchmarks for LLMs show that GLVQ achieves a superior trade-off between accuracy and inference efficiency under extreme compression regimes.

## 2 Background and Related Work

### 2.1 LLM Post-Training Quantization (PTQ)

Recent surveys [19, 21] have emphasized the importance of *post-training quantization* (PTQ) methods in the context of large language models (LLMs), particularly as model scales reach billions of parameters. Classic approaches such as AdaRound [38], BitSplit [56], AdaQuant [24], BRECQ [29], and OBQ [17] have been shown to work well for smaller networks. However, applying these techniques directly to LLMs often incurs prohibitive computational overhead due to their reliance on expensive solvers and per-layer fine-grained adjustments.

In an effort to address scalability challenges, multiple PTQ frameworks have emerged. For instance, GPT3.int8() [9], ZeroQuant [61], and Lut-gemm [41] employed straightforward round-to-nearest (RTN) quantization strategies and introduced granular control to manage the trade-off between model size and accuracy. GPTQ [18] proposed a more data-aware layer-wise optimization to minimize the $\ell_2$ reconstruction error. Follow-up studies [11] argued that 4-bit RTN might be near-optimal for certain classes of networks, but data-aware schemes like GPTQ can exceed this limit without compromising efficiency. Meanwhile, 8-bit quantization of both weights and activations has also been investigated [9, 58, 61], revealing that "outlier features" can substantially degrade performance, and spurring specialized outlier-handling techniques.

Subsequent efforts prioritized addressing such outliers in weight matrices. For example, SpQR [10] maintains a sparse, higher-precision representation of large-magnitude weights to mitigate their outsized influence. Similarly, AWQ [30] employs per-channel scaling to handle channels with large activation magnitudes, while SqueezeLLM [27] adopts diagonal Fisher approximations and K-means clustering to perform non-uniform quantization, selectively preserving critical weights in full precision. A notable result is QuIP [6], which strategically rotates (e.g., via Hadamard transform) and then projects weights onto a lattice to control outliers, enabling 2-bit per-parameter compression with only minor increases in perplexity.

Beyond these developments, recent research introduces vector quantization (VQ) to further enhance LLM compression. Transformer-VQ [31] applies VQ to the Attention key vectors, effectively reducing Attention complexity to linear time. Other VQ-based approaches generally optimize discrete code assignments (often in the form of one-hot vectors) and their associated codebooks [14, 51]. For instance, AQLM [14] proposes an additive quantization strategy that is input-adaptive, while QuIP# [51] leverages a highly symmetric $E_8$ lattice to handle weights that follow (approximately) a spherical sub-Gaussian distribution. GPTVQ [54] interleaves column-wise quantization steps with updates to the unquantized parameters, guided by the Hessian of the per-layer output reconstruction MSE, and then employs SVD and integer quantization for additional codebook compression. PV-Tuning [36] shows how relying on straight-through estimators (STE) alone can be suboptimal, proposing a stage-wise method that separately optimizes continuous parameters (e.g., scales, codebooks) and discrete assignments. QTIP [52] decouples the notion of codebook size from both bitrate and dimensionality through a stateful decoding, thereby facilitating ultra-high-dimensional VQ. NestQuant [46] proposes a novel PTQ scheme for weights and activations that is based on self-similar nested lattices.

### 2.2 Lattice Vector Quantization (LVQ)

Scalar quantization maps each parameter to a discrete code individually. Vector quantization, instead, quantizes a group of parameters as a vector by mapping it to the nearest code vector from a codebook [22]. Lattice-based quantization [8, 20, 15] goes a step further, leveraging a structured grid in the high-dimensional space for efficient quantization. Formally, let $\mathbf{G} \in \mathbb{R}^{d \times d}$ be a *generation matrix*, which defines a full-rank lattice: $\Lambda = \left\{ \mathbf{G}\,\mathbf{z} \,\middle|\, \mathbf{z} \in \mathbb{Z}^d \right\}$. Each column of $\mathbf{G}$ acts as a basis vector in $\mathbb{R}^d$, so any lattice point can be written as an integer combination of these basis vectors.

Given a vector $\mathbf{x} \in \mathbb{R}^d$, the encoding operation aims to find an integer vector $\hat{\mathbf{z}} \in \mathbb{Z}^d$ such that the corresponding lattice point $\mathbf{G}\,\hat{\mathbf{z}}$ is close to $\mathbf{x}$ in the Euclidean sense:

$$\hat{\mathbf{z}} = \arg\min_{\mathbf{z} \in \mathbb{Z}^d} \|\mathbf{x} - \mathbf{G}\,\mathbf{z}\|. \tag{1}$$

In general, solving the nearest–lattice–point problem is both computationally prohibitive for large $d$ and non-differentiable, motivating the use of efficient approximate methods such as *Babai rounding* [1]. Once the integer lattice index $\hat{\mathbf{z}}$ is found and stored, we can decode the quantized vector by multiplying back: $\hat{\mathbf{x}} = \mathbf{G}\,\hat{\mathbf{z}}$, yielding a point on the lattice that approximates the original vector $\mathbf{x}$.

**Learnable LVQ.** Recent studies have shown that learning or adapting the generation matrix $\mathbf{G}$ to a given data distribution can significantly reduce quantization error at a fixed bit budget [26, 63, 60, 59, 64]. This is especially beneficial in low-bit regimes where per-dimension scalar methods often struggle. Additionally, practical implementations of LVQ can incorporate companding or pre/post-transformation steps to further align the data with the lattice geometry [15, 62].

## 2.3 Companding for Quantization

Many neural network weight distributions exhibit high dynamic range and non-uniform behavior (Gaussian or sub-Gaussian), which can amplify quantization errors in the low-magnitude regime. Companding (derived from "compressing" and "expanding") [48, 25, 53] offers a simple, yet effective strategy to reshape these distributions before quantization, thereby reducing distortion in critical regions. Typically, the companding process comprises two stages:

Given a value $x$, , a nonlinear function $F(\cdot)$ is applied before quantization to produce $z = F(x)$, and the inverse function is applied after quantization to obtain the reconstruction $\hat{x} = F^{-1}(\hat{z})$, where $\hat{z}$ denotes the quantized value (e.g., using scalar or lattice-based quantization). This two-step *compress-and-expand* reduces errors near zero, where neural parameters are often concentrated, and avoids over-allocation of code space for very large or very small values.

A well-known example of $F$ is the $\mu$-law transformation [45, 4], originally used in audio signal processing:

$$F_\mu(x) = \mathrm{sgn}(x)\,\frac{\ln\!\big(1 + \mu|x|\big)}{\ln(1 + \mu)}, \tag{2}$$

where $\mu > 0$ is a hyperparameter controlling curvature. This two-step process allocates finer resolution near zero and avoids overuse of code space for large outliers.

## 3 Method

The key idea of **GLVQ** is to endow each weight group with its own lattice codebook, thereby matching the heterogeneous statistics observed across different portions of an LLM. Unfortunately, jointly optimizing both the *size* (bit-width) and the *structure* (lattice basis) of every codebook is combinatorial and intractable. We therefore decompose the problem into two sequential sub-tasks:

1. **Bit allocation.** Given a global bit budget, we first determine the codebook size for each group by assigning an integer bit-width $b_g$. A lightweight heuristic prefers higher precision for groups whose weight statistics exhibit larger dynamic ranges or higher sensitivity, under the constraint $\sum_g b_g \ell_g \leq$ Budget.

2. **Lattice codebook learning.** With $b_g$ fixed (implying a lattice codebook of $2^{b_g}$ points), we reshape every weight group $\mathbf{W}_g \in \mathbb{R}^{m_g \times n_g}$ into a matrix of size $d \times \ell_g$ ($d$ is the lattice dimension). We then learn a group-specific generation matrix $\mathbf{G}_g \in \mathbb{R}^{d \times d}$ and the corresponding integer indices $\mathbf{Z}_g \in \mathbb{Z}^{d \times \ell_g}$ can be obtained via Babai rounding [1]. Hence the quantized weight matrix is

$$\widehat{\mathbf{W}}_g = Q_g(\mathbf{W}_g) = \mathbf{G}_g\,\mathbf{Z}_g.$$

Fixing the size beforehand simplifies the search space and lets the optimizer focus on discovering a lattice *structure* that best fits the local weight distribution. The Babai rounding algorithm does not always return the exact nearest lattice point; nevertheless, its approximation error is formally bounded, as proved in Appendix A.

**Group-specific companding.** To further curb quantization error, we learn a nonlinear companding function $F_g(\cdot)$ per group. Each weight group undergoes

$$\widehat{\mathbf{W}}_g = F_g^{-1}\big(Q_g\big(F_g(\mathbf{W}_g)\big)\big),$$

where $Q_g$ is the lattice quantizer defined above. This adaptive companding allocates finer resolution near weight values that are most influential for the layer's output, yielding markedly lower error than a fixed uniform quantizer.

### 3.1 Salience-Determined Bit Allocation

To achieve optimal bit allocation under strict resource constraints, we adopt the *Salience-Determined Bit Allocation (SDBA)* mechanism proposed in Slim-LLM [23], which dynamically assigns bit-widths to each group based on its importance.

Given the weight matrix $\mathbf{W}$, calibration input $\mathbf{X}$, and a target average bit-width $N$, SDBA solves the following constrained optimization problem:

$$\underset{b_1,\ldots,b_G}{\arg\min}\ D_{\mathrm{KL}}\left(\mathbf{W}\mathbf{X} \,\|\, \widehat{\mathbf{W}}\mathbf{X}\right), \quad \text{s.t.} \quad \frac{1}{G}\sum_{g=1}^{G} b_g = N \quad \text{and} \quad |\mathcal{G}_{N+1}| = |\mathcal{G}_{N-1}|, \tag{3}$$

where $b_g \in \mathbb{Z}^+ \cap \big[N-1,\ N+1\big]$ is the bit-width allocated to group $g$, $\mathcal{G}_{N+1} = \{g \mid b_g = N+1\}$, and $\mathcal{G}_{N-1} = \{g \mid b_g = N-1\}$. This constraint ensures a balanced allocation: the number of groups assigned $N+1$ bits equals those assigned $N-1$ bits, while the majority receive $N$ bits. For example, in 2-bit quantization, the highest-salience groups use 3 bits, an equal number of low-salience groups use 1 bit, and the rest use 2 bits. To solve this efficiently, we leverage Slim-LLM's *double-pointer search algorithm*. For a weight matrix with $m$ output channels and group size $g = 128$, the search space is limited to $[0, m/(2g)]$, requiring only $\mathcal{O}(\log m)$ iterations.

### 3.2 Lattice Codebook Learning

The key innovation behind **GLVQ** is the ability to adaptively learn a specialized lattice structure for each weight group, rather than imposing a fixed, universal lattice (such as the $E_8$ lattice employed by QuIP# [51]). Recent studies, such as Slim-LLM [23], show that even after standard preprocessing, large language models (LLMs) exhibit significant heterogeneity across weight groups, motivating the need for group-specific quantization schemes. Specifically, GLVQ learns a generation matrix $\mathbf{G}_g \in \mathbb{R}^{d \times d}$ for each weight group $g$, where the dimension $d$ (e.g., chosen from $\{8, 16, 32, \ldots\}$) controls both computational complexity and the granularity of quantization. Crucially, the integer lattice indices $\mathbf{Z}_g$ are not directly optimized, but rather determined via Babai rounding, making $\mathbf{G}_g$ the only explicitly learned parameter.

Consider the weight matrix for the $g$-th group, $\mathbf{W}_g \in \mathbb{R}^{m_g \times n_g}$. We reshape it into a matrix with dimension $d \times \ell_g$, by partitioning it into $\ell_g$ contiguous sub-blocks of dimension $d$ and stacking these blocks as columns, such that $d\,\ell_g = m_g n_g$. Quantization then approximates each sub-block using a lattice structure defined by $\mathbf{G}_g$:

$$\widehat{\mathbf{W}}_g = \mathbf{G}_g \mathbf{Z}_g, \tag{4}$$

where $\mathbf{Z}_g \in \mathbb{Z}^{d \times \ell_g}$ holds the integer lattice indices computed by rounding each column of $\mathbf{G}_g^{-1} \mathbf{W}_g$. Given calibration inputs $\mathbf{X} \in \mathbb{R}^{n_g \times N}$ (consisting of $N$ samples), GLVQ minimizes the reconstruction error of the layer outputs after quantization:

$$\mathcal{L}_g = \big\|\mathbf{W}_g \mathbf{X} - \widehat{\mathbf{W}}_g \mathbf{X}\big\|_2^2 = \big\|\mathbf{W}_g \mathbf{X} - \mathbf{G}_g \mathbf{Z}_g \mathbf{X}\big\|_2^2. \tag{5}$$

Due to the discrete nature of $\mathbf{Z}_g$, we solve (5) by alternating between two efficient steps:

1. **Lattice index assignment (fix $\mathbf{G}_g$).** Fixing the lattice basis $\mathbf{G}_g$, we update integer indices using Babai rounding [1]:

$$\mathbf{z}_i = \big\lfloor \mathbf{G}_g^{-1} \mathbf{w}_i \big\rceil, \tag{6}$$

   for each column $\mathbf{w}_i$ of $\mathbf{W}_g$. This assignment is computationally efficient with complexity $\mathcal{O}(d^3)$.

2. **Generation matrix optimization (fix $\mathbf{Z}_g$).** With fixed lattice indices $\mathbf{Z}_g$, we optimize $\mathbf{G}_g$ via gradient descent. The gradient is given by:

$$\nabla_{\mathbf{G}_g}\mathcal{L}_g = -2\big(\mathbf{W}_g\mathbf{X} - \mathbf{G}_g\mathbf{Z}_g\mathbf{X}\big)(\mathbf{Z}_g\mathbf{X})^\top. \tag{7}$$

Spectral normalization is applied after each update to constrain the singular values of $\mathbf{G}_g$ within a stable range $[\sigma_{\min}, \sigma_{\max}]$.

To avoid overfitting to the finite calibration set, GLVQ employs a Frobenius regularization term that penalizes deviations from the initial lattice structure:

$$\mathcal{L}_{\mathrm{reg}} = \lambda\big\|\mathbf{G}_g - \mathbf{G}_g^{(0)}\big\|_2^2, \qquad \lambda = 0.1, \tag{8}$$

where $\mathbf{G}_g^{(0)}$ is initialized using the Cholesky decomposition of the group's covariance matrix, aligning initial lattice directions with the group's principal weight distributions.

### 3.3 Group-Specific Companding

The weight distributions of LLMs are typically heavy-tailed and highly non-uniform; directly quantizing such values wastes many code-points on rare outliers. GLVQ therefore inserts an element-wise *companding* stage before lattice quantization and the corresponding expansion after decoding. Distinct from prior work that uses a single global non-linearity, we learn an independent curvature parameter $\mu_g$ for every group so that the transformation adapts to local statistics.

For group $g$ we adopt the $\mu$-law transformation:

$$F_g(x) = \mathrm{sgn}(x)\,\frac{\ln\big(1 + \mu_g|x|\big)}{\ln(1 + \mu_g)}, \qquad F_g^{-1}(y) = \mathrm{sgn}(y)\,\frac{(1 + \mu_g)^{|y|} - 1}{\mu_g}, \tag{9}$$

where $\mu_g > 0$ controls the compression strength. Applied to the reshaped weights, the overall encode–decode chain is

$$\widetilde{\mathbf{W}}_g = F_g\big(\mathbf{W}_g\big) \quad \rightarrow \quad \mathbf{Z}_g = \big\lfloor \mathbf{G}_g^{-1}\widetilde{\mathbf{W}}_g \big\rceil \quad \rightarrow \quad \widehat{\mathbf{W}}_g = F_g^{-1}\big(\mathbf{G}_g\mathbf{Z}_g\big). \tag{10}$$

Given calibration inputs $\mathbf{X} \in \mathbb{R}^{n_g \times N}$, the reconstruction loss proposed in Eq. 5 can be updated with the new definition of $\mathbf{W}_g$ in Eq. 10, accompanying with a Frobenius penalty on the lattice basis:

$$\mathcal{L}_g = \big\|\mathbf{W}_g\mathbf{X} - \widehat{\mathbf{W}}_g\mathbf{X}\big\|_2^2 + \lambda\|\mathbf{G}_g - \mathbf{G}_0\|_2^2, \qquad \lambda = 0.1. \tag{11}$$

Because both $F_g$ and $F_g^{-1}$ are differentiable in $\mu_g$, we update $\mu_g$ jointly with $\mathbf{G}_g$ via gradient descent, while $\mathbf{Z}_g$ is refreshed by Babai rounding at every iteration.

We initialise

$$\mu_g^{(0)} = 100\,\tanh\big(\kappa_g/10\big), \tag{12}$$

where $\kappa_g$ is the sample kurtosis of $\mathbf{W}_g$, so that heavier-tailed groups start with stronger companding. After each update we project $\mu_g$ onto the practical range $[10, 255]$ to ensure numerical stability.

### 3.4 Quantization Pipeline and Runtime Decoding

We present the full pseudocode for GLVQ in Algorithm 1. Starting from initial values $\mathbf{G}_g^{(0)}$ and $\mu_g^{(0)}$, each iteration (i) reshapes the weight block, applies the group-specific $\mu_g$-law companding, and produces latent vectors $\mathbf{Y}_g$; (ii) quantizes these vectors via Babai rounding to obtain integer codes $\mathbf{Z}_g$; (iii) reconstructs provisional weights $\widehat{\mathbf{W}}_g$ by inverse companding the lattice outputs; and (iv) minimizes a reconstruction loss augmented with a Frobenius penalty on $\mathbf{G}_g$. Gradients update the generation matrix and curvature parameter, while $\mathbf{Z}_g$ is implicitly refreshed at the next rounding step. The loop stops when the relative loss reduction falls below $\varepsilon$, returning the final compact representation $\widehat{\mathbf{W}}_g$ that combines group-specific lattice precision with adaptive companding. This design delivers state-of-the-art accuracy under tight bit budgets while meeting the memory and latency constraints of real-world LLM deployment.

**Offline compression.** After the optimization in Alg. 1, each weight group is represented by (i) an integer-code tensor and (ii) a small set of *side parameters*—a $d\times d$ FP16 generation matrix plus one

**Algorithm 1** Pseudo code of GLVQ algorithm.

---

**Require:** Group weights $\mathbf{W}_g \in \mathbb{R}^{m_g \times n_g}$, calibration inputs $\mathbf{X} \in \mathbb{R}^{n_g \times T}$, initial generation matrix $\mathbf{G}_g^{(0)} \in \mathbb{R}^{d \times d}$, initial companding parameter $\mu_g^{(0)}$

1: $\mathbf{G}_g \leftarrow \mathbf{G}_g^{(0)}, \quad \mu_g \leftarrow \mu_g^{(0)}$
2: **repeat**
3: $\quad \widetilde{\mathbf{W}}_g \leftarrow F_{\mu_g}\big(\text{reshape}(\mathbf{W}_g, [d, \ell])\big)$         {apply companding; $\ell = m_g n_g / d$}
4: $\quad \mathbf{Z}_g \leftarrow \text{round}\big(\mathbf{G}_g^{-1}\widetilde{\mathbf{W}}_g\big)$           {Babai lattice rounding}
5: $\quad \widehat{\mathbf{W}}_g \leftarrow \text{reshape}\big(F_{\mu_g}^{-1}(\mathbf{G}_g\mathbf{Z}_g), [m_g, n_g]\big)$     {inverse compand & reshape}
6: $\quad \mathcal{L}_g \leftarrow \big\|\mathbf{W}_g\mathbf{X} - \widehat{\mathbf{W}}_g\mathbf{X}\big\|_F^2 + \lambda\big\|\mathbf{G}_g - \mathbf{G}_g^{(0)}\big\|_F^2$   {reconstruction + regularizer}
7: $\quad \mathbf{G}_g \leftarrow \mathbf{G}_g - \eta_G \nabla_{\mathbf{G}_g}\mathcal{L}_g$         {update generation matrix}
8: $\quad \mu_g \leftarrow \mu_g - \eta_\mu \nabla_{\mu_g}\mathcal{L}_g$         {update companding parameter}
9: **until** $\big|\mathcal{L}_g^{(t)} - \mathcal{L}_g^{(t-1)}\big|/\mathcal{L}_g^{(t-1)} < \varepsilon$        {convergence check}
10: **return** $\widehat{\mathbf{W}}_g$

---

FP16 scalar $\mu_g$. Because $d$ is modest ($8-32$) and the matrix is shared by *all* codes in the group, the storage overhead is tiny. For the default configuration used in our experiments ($d$=16, 4-bit weights, $m_g$=4096, $n_g$=128), the side information adds only **0.2**% to the weight codes. Aggregated over the entire Llama 2-7B model, this amounts to roughly **2** MB of extra storage on top of a 1.1 GB 4-bit payload—versus 13.4 GB in FP16. A detailed derivation is provided in Appendix B.

**On-the-fly decoding.** During inference we materialise just a handful of sub-blocks, apply $\widehat{\mathbf{w}} = F_g^{-1}\big(\mathbf{G}_g\mathbf{z}\big)$ and release the data immediately after use. This streaming strategy cuts peak activation-plus-weight memory by $> \mathbf{10}\times$ versus the common practice of pre-decompressing an entire layer. The additional computation is modest—$d^2+d$ multiplies per sub-block—and increases end-to-end latency by only 2–3% relative to a standard 4-bit uniform PTQ baseline.

## 4 Experiments

In this section, we evaluate the performance of our proposed grouped Lattice vector quantization (GLVQ) framework on the Llama 1 and 2 models [50], which range from 7 billion to 70 billion parameters. To ensure a fair comparison, we follow the experimental setup used in QuIP#. Specifically, we evaluate perplexity on the Wikitext-2 [37] and C4 [44] datasets, utilizing context lengths of 2048 for Llama 1 and 4096 for Llama 2 models. For zero-shot tasks, we use the LM Eval framework to measure accuracy on tasks such as ARC, PIQA, and the Winograd Schema Challenge (Wino). We report both perplexity and zero-shot accuracy for all methods to offer a comprehensive evaluation. We implement our method using PyTorch [42] and CUDA [39], with all experiments conducted on NVIDIA A100 GPUs. For timing experiments, we use an NVIDIA RTX 4090 GPU. We adopt 4M tokens from the RedPajama 1T dataset [57] as the calibration sequences in our experiments. We implement two variants of our model with lattice dimensions $d = 8$ and $d = 32$, referred to as **GLVQ-8D** and **GLVQ-32D**, respectively. For baselines, we compare our method against several state-of-the-art PTQ approaches, including OmniQuant [47], AWQ [30] QuIP# [51], AQLM [14] and QTIP [52].

### 4.1 Perplexity Results on Llama Models

Our evaluation focuses on perplexity over the Wikitext-2 [37] and C4 [44] datasets, using a context length of 2048 for Llama 1 and 4096 for Llama 2. Results are reported in Table 1. Across all settings, GLVQ consistently achieves lower perplexity than existing methods, particularly in low-bit regimes. For example, under the 2-bit quantization regime on Llama 2-70B, GLVQ-32D achieves a perplexity of 3.36 on Wikitext-2, significantly lower than QTIP (3.78) and QuIP# (3.91), highlighting that the performance gap becomes most pronounced under extreme compression levels. We also observe consistent improvements across model scales and datasets. On Llama 1-13B, GLVQ-32D improves upon QuIP# by 0.41 perplexity points (5.38 vs. 5.79) at 2-bit on Wikitext-2, and 0.53 points (6.95 vs. 7.48) on C4. These results validate the robustness of our method across both small and large Llama variants. Notably, the performance gain is more pronounced as the bit-width decreases, highlighting the advantage of GLVQ's group-specific lattice codebooks and companding

Table 1: Perplexity (↓) of competing methods on Wikitext2 and C4 for Llama 1 and 2. All results use a context length of 2048 for Llama 1 and 4096 for Llama 2.

| Method | Bits | Wikitext 2 | | | | | | | C4 | | | | | | |
|---|---|---|---|---|---|---|---|---|---|---|---|---|---|---|---|
| | | 1-7 | 1-13 | 1-30 | 1-65 | 2-7 | 2-13 | 2-70 | 1-7 | 1-13 | 1-30 | 1-65 | 2-7 | 2-13 | 2-70 |
| Fp16 | 16 | 5.68 | 5.09 | 4.10 | 3.53 | 5.12 | 4.57 | 3.12 | 7.08 | 6.61 | 5.98 | 5.62 | 6.63 | 6.05 | 4.97 |
| OmniQ | 2 | 15.5 | 13.2 | 8.71 | 7.58 | – | – | – | 24.9 | 18.3 | 13.9 | 10.8 | – | – | – |
| QuIP# | 2 | 6.86 | 5.97 | 5.02 | 4.36 | 6.19 | 5.35 | 3.91 | 8.36 | 7.48 | 6.71 | 6.19 | 8.16 | 7.20 | 5.71 |
| QTIP | 2 | 6.52 | 5.80 | 4.83 | 4.21 | 5.91 | 5.26 | 3.78 | 7.99 | 7.31 | 6.56 | 6.08 | 7.76 | 6.99 | 5.56 |
| **GLVQ-8D** | 2 | **6.28** | **5.64** | **4.57** | **4.01** | **5.69** | **5.02** | **3.62** | **7.83** | **7.17** | **6.48** | **5.94** | **7.33** | **6.56** | **5.25** |
| **GLVQ-32D** | 2 | **6.00** | **5.38** | **4.32** | **3.81** | **5.41** | **4.80** | **3.36** | **7.61** | **6.95** | **6.22** | **5.80** | **7.14** | **6.33** | **5.04** |

Table 2: Zero-shot accuracy (*acc* in LM Eval, not *acc_norm*) for Llama 2 models.

| | 2-70 | | | | 2-13 | | | | 2-7 | | | |
|---|---|---|---|---|---|---|---|---|---|---|---|---|
| Method | Bits | ArcC | ArcE | PIQA | WINO | Bits | ArcC | ArcE | PIQA | WINO | Bits | ArcC | ArcE | PIQA | WINO |
| FP16 | 16 | 51.1 | 77.7 | 81.1 | 77.0 | 16 | 45.6 | 73.3 | 73.5 | 69.6 | 16 | 40.0 | 69.3 | 78.5 | 67.3 |
| OmniQ | 4 | 49.8 | 77.9 | 80.7 | 75.8 | 4 | 43.1 | 70.2 | 78.4 | 67.8 | 4 | 37.9 | 67.8 | 77.1 | 67.0 |
| QuIP | 4 | 47.0 | 74.3 | 80.3 | 76.0 | 4 | 44.9 | 73.3 | 79.0 | 69.7 | 4 | – | – | – | – |
| AQLM | 4 | 51.0 | 78.1 | 81.4 | 76.9 | 4 | 43.9 | 72.2 | 78.6 | 70.4 | 4 | 40.3 | 68.9 | 77.7 | 67.3 |
| QuIP# | 4 | 50.6 | **78.1** | 81.4 | 77.1 | 4 | 45.5 | 73.9 | 78.9 | 69.9 | 4 | 40.5 | 69.1 | 78.4 | 67.6 |
| QTIP | 4 | 50.0 | 77.6 | 81.5 | 77.0 | 4 | 45.9 | 73.2 | 78.6 | 69.9 | 4 | 40.4 | 69.2 | 78.5 | 67.4 |
| **GLVQ-8D** | 4 | **51.2** | 78.0 | **81.6** | **77.3** | 4 | **46.0** | **74.0** | **79.2** | **70.5** | 4 | **40.7** | **69.5** | **78.6** | **67.8** |
| OmniQ | 3 | 47.6 | 75.7 | 79.7 | 73.5 | 3 | 42.0 | 69.0 | 77.7 | 65.9 | 3 | 35.3 | 62.6 | 73.6 | 63.6 |
| QuIP | 3 | 46.3 | 73.2 | 80.0 | 74.6 | 3 | 41.5 | 70.4 | 76.9 | 69.9 | 3 | – | – | – | – |
| AQLM | 3 | 50.0 | 77.6 | 81.3 | **77.2** | 3 | 43.6 | 73.5 | 77.8 | 67.6 | 3 | 38.7 | 67.8 | 76.6 | 68.4 |
| QuIP# | 3 | **50.9** | 77.7 | 81.4 | 76.4 | 3 | 44.0 | 72.5 | 78.4 | 69.1 | 3 | 39.2 | 68.4 | 77.3 | 66.5 |
| QTIP | 3 | 49.8 | 77.5 | 81.2 | 76.3 | 3 | 43.6 | 72.5 | 78.2 | 69.5 | 3 | 39.8 | 69.7 | 78.0 | 66.8 |
| **GLVQ-8D** | 3 | 50.8 | **78.0** | **81.6** | 77.1 | 3 | **46.5** | **74.2** | **79.3** | **71.0** | 3 | **40.9** | **69.8** | **78.8** | **67.9** |
| OmniQ | 2 | 28.7 | 55.4 | 68.8 | 53.2 | 2 | 23.0 | 44.4 | 62.6 | 52.6 | 2 | 21.6 | 35.2 | 57.5 | 51.5 |
| QuIP | 2 | 34.0 | 62.2 | 74.8 | 67.5 | 2 | 23.5 | 45.2 | 62.0 | 52.8 | 2 | 19.4 | 26.0 | 54.6 | 51.8 |
| AQLM | 2 | 47.9 | **77.7** | 80.4 | 75.9 | 2 | 38.5 | 67.0 | 75.1 | 69.5 | 2 | 33.6 | 62.8 | 73.5 | 64.6 |
| QuIP# | 2 | 48.7 | 77.3 | 80.3 | 75.9 | 2 | 39.5 | 69.3 | 77.3 | 67.7 | 2 | 34.6 | 64.6 | 75.1 | 64.9 |
| QTIP | 2 | 48.1 | 76.9 | 80.1 | **76.5** | 2 | 39.2 | 70.6 | 77.8 | 71.0 | 2 | 35.3 | 63.9 | 75.3 | 66.7 |
| **GLVQ-8D** | 2 | **49.0** | 77.5 | **80.5** | 76.1 | 2 | **40.0** | 70.8 | 78.0 | 71.3 | 2 | **35.5** | 64.5 | 75.5 | 67.2 |

functions. The combination of adaptive vector quantization and nonlinear transformation helps mitigate the quantization error that typically worsens at extreme compression levels. Furthermore, GLVQ-32D consistently outperforms GLVQ-8D, demonstrating that larger lattice dimensions improve quantization fidelity by offering a more expressive codebook structure. This supports our design choice of learning group-specific lattices with adjustable resolution.

## 4.2 Zero-Shot Accuracy on Llama Models

We report zero-shot accuracy of GLVQ and baseline methods on Llama-2 models under 4-bit, 3-bit and 2-bit quantization across four tasks from LM Eval: ARC-Challenge, ARC-Easy, PIQA, and Winogrande. Results are summarized in Table 2. GLVQ-8D consistently matches or outperforms existing methods across all bit-widths and model sizes. At 4-bit precision, GLVQ-8D achieves the highest accuracy on most tasks, e.g., 51.2% on ARC-Challenge and 81.6% on PIQA (2-70B), slightly surpassing QTIP and QuIP#. As the bit-width decreases, the performance gap becomes more evident. At 2-bit precision, GLVQ maintains strong performance—e.g., 40.0% on ARC-Challenge and 78.0% on PIQA (2-13B)—outperforming QuIP# (39.5%, 77.3%) and QTIP (39.2%, 77.8%). On the smallest model (2-7B), GLVQ-8D again leads with the highest scores on all tasks, demonstrating its robustness under extreme compression. These results confirm that GLVQ preserves downstream task accuracy more effectively than existing vector quantization methods, especially in low-bit regimes.

## 4.3 Fractional and Sub-2-Bit Quantization

To assess GLVQ in more extreme compression regimes, we further evaluate its performance under fractional and ultra-low bit-width settings (1.5 and 1.0 average bits per weight), using the same protocol as in the main experiments. Fractional rates are naturally supported in GLVQ via group-wise bit allocation: each group is assigned an integer bit-width, and the global rate is the arithmetic mean across groups, allowing any rational target (e.g., 1.5 bit by mixing 1-bit and 2-bit groups). No architectural or algorithmic modification is required to support such mixtures.

Table 3: Perplexity (↓) of fractional and sub-2-bit quantization on Wikitext2 (Llama-2). GLVQ achieves strong performance at 1.5-bit and remains competitive even at 1.0-bit.

| Method | 7B | | | 13B | | | 70B | | |
|---|---|---|---|---|---|---|---|---|---|
| | Bits | PPL | Δ to GLVQ | Bits | PPL | Δ | Bits | PPL | Δ |
| BiLLM | 1.08 | 32.48 | +24.65 | 1.10 | 16.77 | +10.66 | 1.08 | 8.41 | +2.30 |
| OneBit | 1.00 | 9.73 | +1.90 | 1.00 | 8.76 | +1.17 | – | – | – |
| PV-Tuning | 1.02 | 8.28 | +0.45 | 0.97 | 7.96 | +0.37 | 1.00 | 6.50 | +0.39 |
| **GLVQ (ours)** | 1.00 | 7.83 | 0.00 | 1.00 | 7.59 | 0.00 | 1.00 | 6.11 | 0.00 |
| PB-LLM | 1.70 | 69.20 | +62.19 | 1.70 | 151.09 | +144.98 | 1.70 | 28.37 | +23.38 |
| PV-Tuning | 1.58 | 7.32 | +0.31 | 1.37 | 6.65 | +0.54 | 1.14 | 5.52 | +0.53 |
| **GLVQ (ours)** | 1.50 | 7.01 | 0.00 | 1.50 | 6.11 | 0.00 | 1.50 | 4.99 | 0.00 |

Table 4: Inference throughput (TOK/s), MATVEC MEM BW (GB/S), and perplexity on WikiText2 dataset tested on a NVIDIA RTX 4090 for various methods on 2-bit Llama 2-7B and 2-70B models. OOM indicates that the method ran out of memory on the tested model.

| Method | 2-7B | | | 2-70B | | |
|---|---|---|---|---|---|---|
| | TOK/S | MEM BW | Perplexity (↓) | TOK/S | MEM BW | Perplexity (↓) |
| Original Model | 33.1 | – | 7.0 | OOM | OOM | – |
| **Scalar Quantization** | | | | | | |
| OmniQuant | 125.0 | – | 37.4 | 35.1 | – | 7.81 |
| GPTVQ-1D | 143.5 | – | 10.1 | 40.2 | – | 4.82 |
| **Vector Quantization** | | | | | | |
| GPTVQ(4D) | 24.0 | – | 7.22 | 10.4 | – | 4.39 |
| AQLM | 20.6 | – | 6.59 | 8.27 | – | 3.94 |
| QUIP# | 106.3 | 697 GB/s | 6.66 | 25.9 | 867 GB/s | 4.16 |
| QTIP | 105.2 | 628 GB/s | 5.91 | 25.8 | 840 GB/s | 3.78 |
| GLVQ-8D-u | 102.3 | 615 GB/s | 5.87 | 24.9 | 836 GB/s | 3.72 |
| GLVQ-32D-u | 100.2 | 608 GB/s | 5.55 | 24.1 | 831 GB/s | 3.49 |
| GLVQ-8D | 88.4 | 544 GB/s | 5.69 | 20.1 | 721 GB/s | 3.62 |
| GLVQ-32D | 82.0 | 521 GB/s | 5.41 | 18.7 | 702 GB/s | 3.36 |

Table 3 reports results on WikiText2 for Llama-2 at 1.5 bit and 1.0 bit. At 1.5 bit, GLVQ achieves perplexities of 7.01 (7B), 6.11 (13B), and 4.99 (70B), outperforming PV-Tuning at matched rates and significantly surpassing PB-LLM. Notably, GLVQ approaches the performance of 2-bit baselines despite using substantially fewer bits. Even at the extremely aggressive 1.0 bit regime, GLVQ remains competitive: it attains 7.83 (7B), 7.59 (13B), and 6.11 (70B), outperforming BiLLM and OneBit by large margins on 7B and 13B and closely matching PV-Tuning at similar budgets.

These findings highlight the robustness of GLVQ's group-specific lattice design and salience-driven allocation under extreme compression, where competing methods degrade substantially. Unlike approaches that depend on retraining or fine-tuning, GLVQ preserves strong performance even in the ultra-low-rate regime, making it a practical solution for deployment in severe memory- and energy-constrained environments.

## 4.4 Inference Throughput

We measure inference throughput on a single NVIDIA RTX 4090 GPU, using a batch size of 1 and a 2-bit quantization setup for all competing methods. The results are summarized in Table 4. We compare both uniform-precision and mixed-precision variants of GLVQ. The uniform versions (denoted as GLVQ-u) do not use salience-drive bit allocation and instead apply a fixed bit-width across all groups. Despite this simplification, GLVQ-8D-u and GLVQ-32D-u achieve throughput on par with QTIP (e.g., 100.2 vs. 105.2 TOK/s on Llama 2-7B), while consistently delivering lower perplexity (e.g., 5.55 vs. 5.91), demonstrating the effectiveness of group-specific lattice quantization and companding. The full GLVQ variants, which leverage salience-determined bit allocation, achieve even better perplexity (e.g., 5.41 with GLVQ-32D vs. 5.91 with QTIP) at the cost of a moderate decrease in speed (e.g., 82.0 vs. 105.2 TOK/s). The reduction in memory bandwidth usage also confirms improved efficiency (e.g., 521 GB/s vs. 628 GB/s). These results highlight that GLVQ offers a flexible trade-off between accuracy and efficiency, and its mixed-precision configuration is particularly advantageous for accuracy-sensitive scenarios such as language generation or summarization.

### 4.5 Ablation Studies

To quantify the contribution of each design component in GLVQ, we perform controlled ablations and examine their effects on perplexity. All corresponding results are reported in Appendix Tables 6–13.

**Bit allocation.** We replace the salience-driven bit allocation with a uniform bit-width across all groups (Appendix D, Table 6). This forces all groups to share the same precision regardless of their sensitivity. Perplexity increases consistently across model scales and bit-widths; for example, on Llama 1–13B at 2-bit it rises from 5.64 to 5.79, and on Llama 2–70B from 3.62 to 3.72. The gap persists at 3-bit and remains visible even at 4-bit, where the precision constraint is relatively loose. These results confirm that group-aware precision allocation is essential for preserving accuracy when operating under tight bit budgets.

**Group-specific lattice codebook.** We then enforce a single shared lattice basis across all groups instead of learning group-specific bases (Appendix E, Table 7). This substitution consistently harms performance, especially under 2-bit quantization (e.g., Llama 1–13B increases from 5.64 to 5.89; Llama 2–70B from 3.62 to 3.88), and the degradation persists at 3- and 4-bit. These findings indicate that different weight groups exhibit distinct geometric statistics, and a shared codebook cannot align to them simultaneously. Group-specific lattice learning is therefore necessary to realize the full benefit of lattice quantization.

**Group-specific companding.** Next, we remove the group-wise $\mu$-law companding and apply a fixed global transformation to all groups (Appendix F, Table 8). Perplexity increases across all settings, with the strongest effect again observed at low-bit precision (e.g., Llama 1–13B at 2-bit: 5.64 to 5.87; Llama 2–70B: 3.62 to 3.88). This validates the hypothesis that heavy-tailed and skewed weight distributions must be linearized at the group level to reduce quantization error.

**Group size.** We ablate the number of columns per group using {32, 64, 128, 256, 512} (Appendix G, Tables 9, 10). Very small groups (32/64) yield slightly lower perplexity but incur significant codebook/storage overhead, whereas very large groups (256/512) compromise accuracy by averaging out distributional differences within the group. A moderate group size of 128 consistently provides the best balance between the adaptation power and additional overhead and is therefore adopted as the default option in the proposed GLVQ.

**Calibration-set size.** We vary the number of calibration tokens from 100K to 8M (Appendix H, Table 11). Perplexity improves steadily up to about 4M tokens and then saturates; for instance, on Llama 1–13B (2-bit) perplexity is 5.64 at 4M and 5.65 at 8M. Importantly, GLVQ remains competitive even with 500K or 1M tokens, and degradation is gradual even at 100K. This demonstrates both the data efficiency and the robustness of the method in low-calibration regimes.

**Babai rounding vs. GCD.** Finally, we replace Babai rounding with greedy coordinate descent (GCD) for index assignment (Appendix I, Tables 12, 13). GCD consistently yields worse perplexity; for example, on Llama 1–13B at 2-bit the perplexity rises from 5.64 (Babai) to 5.92 (GCD). Zero-shot results on Llama 2 benchmarks show the same pattern. This confirms that Babai rounding offers both better convergence stability and superior final accuracy, and should be preferred for lattice-based quantization.

## 5 Conclusion

We have introduced a novel approach, grouped lattice vector quantization (GLVQ), for low-bit mixed-precision quantization of Large Language Models (LLMs). By integrating group-wise bit allocation, lattice codebook learning, and companding transformations, GLVQ achieves superior rate-distortion performance compared to scalar or fixed-precision baselines. Our experimental results demonstrate the effectiveness of GLVQ on both language modeling and summarization tasks, highlighting its potential as a practical solution for deploying LLMs in resource-constrained environments. Future research will focus on exploring hierarchical grouping strategies and designing hardware-efficient lattice codebooks.

**Limitations & future work.** GLVQ still assumes a fixed column partition, which may under-utilize layer-wise sensitivity structure; data-driven grouping may improve rate–distortion trade-offs. The current work also compresses only weights; extending lattice-based quantization to activations would unlock further memory and energy savings but is complicated by dynamic activation statistics.

## Acknowledgements

This research is supported by the RIE2025 Industry Alignment Fund – Industry Collaboration Projects (IAF-ICP) (Award I2301E0026), administered by A*STAR, as well as supported by Alibaba Group and NTU Singapore through Alibaba-NTU Global e-Sustainability CorpLab (ANGEL). This work is also supported in part by the National Natural Science Foundation of China (No.62301313).

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

# Technical Appendices and Supplementary Material

## A  A Brief Theoretical Derivation of Babai Rounding's Error Bound

### A.1  Algorithm Description

Babai's rounding algorithm is a classical method for finding an approximate closest lattice point to a given target vector. Let

$$B = [\mathbf{b}_1, \ldots, \mathbf{b}_n] \in \mathbb{R}^{m \times n}$$

be a lattice basis and $\mathbf{t} \in \mathbb{R}^m$ be the target vector. The algorithm proceeds in three steps:

1. Compute the real coordinate vector: $\mathbf{x} = B^{-1}\mathbf{t}$. This step expresses the target vector $\mathbf{t}$ in terms of the lattice basis $B$.

2. Round each coordinate: $c_i = \lfloor x_i + 0.5 \rfloor$. Here, each coordinate $x_i$ is rounded to the nearest integer $c_i$.

3. Return the lattice point: $\mathbf{v} = B\mathbf{c}$. This gives the approximate closest lattice point.

The error vector $\mathbf{e}$ is defined as the difference between the target vector and the approximate lattice point:

$$\mathbf{e} = \mathbf{t} - \mathbf{v} = B(\mathbf{x} - \mathbf{c}) = B\boldsymbol{\delta}, \tag{13}$$

where $\boldsymbol{\delta} = (\delta_1, \ldots, \delta_n)^\top$ and $|\delta_i| \leq \frac{1}{2}$ due to rounding.

### A.2  Gram-Schmidt Orthogonalization

To analyze the error, we first orthogonalize the lattice basis $B$ using the Gram-Schmidt process. Let

$$B^* = [\mathbf{b}_1^*, \ldots, \mathbf{b}_n^*]$$

be the orthogonalized basis, where each $\mathbf{b}_i$ can be expressed as:

$$\mathbf{b}_i = \mathbf{b}_i^* + \sum_{j=1}^{i-1} \mu_{j,i} \mathbf{b}_j^*. \tag{14}$$

Here,

$$\mu_{j,i} = \frac{\langle \mathbf{b}_i, \mathbf{b}_j^* \rangle}{\|\mathbf{b}_j^*\|^2}$$

represents the projection coefficient of $\mathbf{b}_i$ onto $\mathbf{b}_j^*$. This decomposition ensures that $\mathbf{b}_i^*$ is orthogonal to all previous $\mathbf{b}_j^*$ for $j < i$.

### A.3  Error Vector Decomposition

Next, we decompose the error vector $\mathbf{e}$ in terms of the orthogonal basis $B^*$. Starting from the definition of $\mathbf{e}$, we substitute the expression for $\mathbf{b}_i$:

$$\mathbf{e} = \sum_{i=1}^{n} \delta_i \mathbf{b}_i = \sum_{i=1}^{n} \delta_i \left( \mathbf{b}_i^* + \sum_{j=1}^{i-1} \mu_{j,i} \mathbf{b}_j^* \right). \tag{15}$$

By rearranging the terms, we can rewrite $\mathbf{e}$ as:

$$\mathbf{e} = \sum_{j=1}^{n} \left( \delta_j + \sum_{i=j+1}^{n} \delta_i \mu_{j,i} \right) \mathbf{b}_j^*. \tag{16}$$

Let $\alpha_j = \delta_j + \sum_{i=j+1}^{n} \delta_i \mu_{j,i}$. Then,

$$\mathbf{e} = \sum_{j=1}^{n} \alpha_j \mathbf{b}_j^*. \tag{17}$$

### A.4 Norm Calculation

Since the vectors $\mathbf{b}_j^*$ are orthogonal, the squared norm of $\mathbf{e}$ is:

$$\|\mathbf{e}\|^2 = \sum_{j=1}^{n} \alpha_j^2 \|\mathbf{b}_j^*\|^2. \tag{18}$$

### A.5 Coefficient Bound

To bound $\|\mathbf{e}\|$, we first derive an upper bound for each coefficient $\alpha_j$.

**Lemma A.1.** *For any $1 \leq j \leq n$, the coefficient $\alpha_j$ satisfies:*

$$|\alpha_j| \leq \frac{1}{2} \left( 1 + \sum_{i=j+1}^{n} |\mu_{j,i}| \right). \tag{19}$$

*Proof.* Using the triangle inequality and the fact that $|\delta_i| \leq \frac{1}{2}$ for all $i$, we have:

$$|\alpha_j| \leq |\delta_j| + \sum_{i=j+1}^{n} |\delta_i||\mu_{j,i}| \leq \frac{1}{2} + \frac{1}{2} \sum_{i=j+1}^{n} |\mu_{j,i}|. \tag{20}$$

### A.6 LLL-Reduced Basis Case

When the basis $B$ is LLL-reduced (with parameter $\delta = 3/4$), the projection coefficients satisfy:

$$|\mu_{j,i}| \leq \frac{1}{2} \quad \forall j < i. \tag{21}$$

Thus, for each $j$,

$$\sum_{i=j+1}^{n} |\mu_{j,i}| \leq \frac{n-j}{2}. \tag{22}$$

### A.7 Final Error Bound

Combining the above results, we obtain:

$$\|\mathbf{e}\|^2 \leq \sum_{j=1}^{n} \left( \frac{1}{2} \left( 1 + \sum_{i=j+1}^{n} |\mu_{j,i}| \right) \right)^2 \|\mathbf{b}_j^*\|^2. \tag{23}$$

For an LLL-reduced basis, using the bound $\sum_{i=j+1}^{n} |\mu_{j,i}| \leq \frac{n-j}{2}$, we have:

$$\|\mathbf{e}\|^2 \leq \frac{1}{4} \sum_{j=1}^{n} \left( 1 + \frac{n-j}{2} \right)^2 \|\mathbf{b}_j^*\|^2. \tag{24}$$

Taking the square root of both sides, the final error bound is:

$$\|\mathbf{e}\| \leq \frac{1}{2} \sqrt{\sum_{j=1}^{n} \left( 1 + \frac{n-j}{2} \right)^2 \|\mathbf{b}_j^*\|^2}. \tag{25}$$

# B    Storage Overhead Analysis

For each group $g$ we store the integer codes for $m_g n_g$ weights ($b_g$ bits per weight) and a small amount of side information: a $d \times d$ FP16 generation matrix plus one FP16 scalar,

$$\text{Bytes}_{\text{wt}}(g) = \frac{m_g n_g b_g}{8}, \quad \text{Bytes}_{\text{sid}}(g) = 2d^2 + 2. \tag{26}$$

The overhead ratio is therefore

$$\text{OH}_g = \frac{\text{Bytes}_{\text{sid}}(g)}{\text{Bytes}_{\text{wt}}(g)} = \frac{16d^2 + 16}{m_g n_g b_g}. \tag{27}$$

**Representative cases.**    Table 5 reports the side-information overhead for a fixed row-count $m_g = 4096$ and two typical column widths ($n_g = 128, 256$), evaluated at lattice dimensions $d \in \{8, 16, 32\}$ and bit-widths $b_g \in \{2, 3, 4\}$. Even in the most demanding configuration (2-bit, $d = 32$, $n_g = 128$), the FP16 lattice parameters add only $\approx 1.6\%$ to the weight codes, while the default setting used in our main experiments ($d = 16$, $n_g = 128$, 4-bit) incurs less than $0.2\%$ overhead—confirming that the side information is negligible across all practical scenarios.

**Observation.**    Even with the largest lattice ($d{=}32$) and the most aggressive 2-bit setting, the side-information overhead is below $1.6\%$. For the default configuration used in our main experiments ($d{=}16$, $n_g{=}128$, $b_g{=}4$), the overhead is under $0.2\%$, confirming that the FP16 lattice parameters contribute a negligible addition to the overall model size.

Table 5: Per-group storage overhead (%) with explicit inclusion of the row count $m_g$. Results are reported for $m_g = 4096$, column widths $n_g \in \{128, 256\}$, lattice dimensions $d \in \{8, 16, 32\}$, and bit-widths $b_g \in \{2, 3, 4\}$.

| $d$ | $m_g$ | $n_g$ | Overhead (%)[†] |
|---|---|---|---|
|  |  |  | $b_g = 2$ / 3 / 4 |
| 8 | 4096 | 128 | 0.10 / 0.07 / 0.05 |
| 8 | 4096 | 256 | 0.05 / 0.03 / 0.02 |
| 16 | 4096 | 128 | 0.39 / 0.26 / 0.20 |
| 16 | 4096 | 256 | 0.20 / 0.13 / 0.10 |
| 32 | 4096 | 128 | 1.56 / 1.04 / 0.78 |
| 32 | 4096 | 256 | 0.78 / 0.52 / 0.39 |

[†] Values computed via Eq. (27).

# C    Broader Impact

**Potential positive societal impacts.**: (i) **Energy and carbon reduction.** By enabling 2–4 bit deployment of large language models (LLMs) on commodity GPUs or edge SoCs, our method lowers inference energy per token and reduces the overall carbon footprint of AI services. (ii) **Edge privacy and accessibility.** Supporting on-device inference allows sensitive user data to remain local, strengthening privacy guarantees, and makes advanced language capabilities available on low-resource or offline devices, fostering digital inclusion. (iii) **Cost-effective R&D.** Lower hardware requirements democratise experimentation for academia, small enterprises, and regions with limited compute infrastructure, broadening participation in AI research and application development.

**Potential negative societal impacts.**: (i) **Easier large-scale misuse.** Compressing powerful LLMs lowers the barrier to deploying them in disinformation, spam, or malicious content generation pipelines. (ii) **Amplified bias under quantisation.** Quantisation can introduce distribution shift that disproportionately affects under-represented dialects or demographic groups, potentially worsening fairness metrics if not carefully audited. (iii) **Expanded surveillance capabilities.** Running compact LLMs on edge cameras or mobile devices may increase ubiquitous monitoring and privacy intrusion when deployed without appropriate safeguards.

## D Ablation Study of Bit Allocation

We assess the impact of group-wise bit allocation by disabling it and assigning a uniform bit-width across all groups. This prevents GLVQ from adapting to the varying importance of different weight groups, resulting in a measurable drop in accuracy. As shown in Table 3 (see Appendix), removing bit allocation consistently increases perplexity across Llama models and bit-widths. For instance, on Llama 1-13B with 2-bit quantization, perplexity increases from 5.64 to 5.79. The degradation becomes more evident at lower bit-widths, where precision budgets are more constrained. These results confirm that salience-driven bit allocation plays a critical role in preserving model accuracy under aggressive compression.

Table 6: Ablation study on salience-determined bit allocation. We compare GLVQ with and without bit allocation across multiple Llama models on Wikitext2 (perplexity $\downarrow$).

| Method | 1-7B | 1-13B | 1-30B | 1-65B | 2-7B | 2-13B | 2-70B |
|---|---|---|---|---|---|---|---|
| GLVQ-8D w/ bit allocation (2-bit) | **6.28** | **5.64** | **4.57** | **4.01** | **5.69** | **5.02** | **3.62** |
| GLVQ-8D w/o bit allocation (2-bit) | 6.44 | 5.79 | 4.65 | 4.10 | 5.87 | 5.19 | 3.72 |
| GLVQ-8D w/ bit allocation (3-bit) | **5.72** | **5.10** | **4.11** | **3.56** | **5.13** | **4.48** | **3.14** |
| GLVQ-8D w/o bit allocation (3-bit) | 5.81 | 5.18 | 4.17 | 3.61 | 5.21 | 4.55 | 3.21 |
| GLVQ-8D w/ bit allocation (4-bit) | **5.70** | **5.08** | **4.09** | **3.52** | **5.01** | **4.44** | **3.02** |
| GLVQ-8D w/o bit allocation (4-bit) | 5.74 | 5.13 | 4.14 | 3.56 | 5.08 | 4.49 | 3.08 |

## E Ablation Study of Group-Specific Lattice Quantization

We evaluate the influence of group-specific lattice learning by replacing each group's learned generation matrix with a *fixed* lattice basis (shared across all groups). As summarised in Table 7, removing adaptive lattice optimisation consistently increases perplexity for every model size and bit-width. The degradation is most evident in the 2-bit setting, where the fixed lattice cannot align with group-wise statistics (e.g., Llama 1-13B rises from 5.64 to 5.89). Even at 4 bits, a shared lattice still hurts accuracy, confirming that the learned, group-specific basis is an essential component of GLVQ.

Table 7: Ablation study on group-specific lattice codebook learning. We compare GLVQ with learned vs. fixed lattice bases across multiple Llama models on WikiText-2 (perplexity $\downarrow$).

| Method | 1-7B | 1-13B | 1-30B | 1-65B | 2-7B | 2-13B | 2-70B |
|---|---|---|---|---|---|---|---|
| GLVQ-8D w/ adaptive lattice (2-bit) | **6.28** | **5.64** | **4.57** | **4.01** | **5.69** | **5.02** | **3.62** |
| GLVQ-8D w/ fixed lattice (2-bit) | 6.53 | 5.89 | 4.82 | 4.26 | 5.95 | 5.28 | 3.88 |
| GLVQ-8D w/ adaptive lattice (3-bit) | **5.72** | **5.10** | **4.11** | **3.56** | **5.13** | **4.48** | **3.14** |
| GLVQ-8D w/ fixed lattice (3-bit) | 5.87 | 5.25 | 4.26 | 3.70 | 5.27 | 4.62 | 3.28 |
| GLVQ-8D w/ adaptive lattice (4-bit) | **5.70** | **5.08** | **4.09** | **3.52** | **5.01** | **4.44** | **3.02** |
| GLVQ-8D w/ fixed lattice (4-bit) | 5.78 | 5.16 | 4.17 | 3.60 | 5.09 | 4.52 | 3.10 |

## F Ablation Study of Group-Specific Companding

We also examine the contribution of group-specific $\mu$-law companding by replacing it with a fixed transformation across all groups. As shown in Table 4 (see Appendix), removing this component significantly degrades performance, particularly in low-bit settings where weight distributions are more difficult to approximate with uniform quantizers. For example, at 2-bit quantization on Llama 1-13B, turning off companding raises the perplexity from 5.64 to 5.87. This confirms that companding helps mitigate quantization error by better accommodating heavy-tailed weight distributions. Overall, these ablation results validate the necessity of group-specific companding as a core component of the proposed GLVQ.

Table 8: Ablation study on group-specific $\mu$-law companding. We compare GLVQ with and without companding across multiple Llama models on Wikitext2 (perplexity $\downarrow$).

| Method | 1-7B | 1-13B | 1-30B | 1-65B | 2-7B | 2-13B | 2-70B |
|---|---|---|---|---|---|---|---|
| GLVQ-8D w/ companding (2-bit) | **6.28** | **5.64** | **4.57** | **4.01** | **5.69** | **5.02** | **3.62** |
| GLVQ-8D w/o companding (2-bit) | 6.58 | 5.87 | 4.71 | 4.18 | 5.97 | 5.23 | 3.88 |
| GLVQ-8D w/ companding (3-bit) | **5.72** | **5.10** | **4.11** | **3.56** | **5.13** | **4.48** | **3.14** |
| GLVQ-8D w/o companding (3-bit) | 5.84 | 5.20 | 4.19 | 3.63 | 5.23 | 4.58 | 3.23 |
| GLVQ-8D w/ companding (4-bit) | **5.70** | **5.08** | **4.09** | **3.52** | **5.01** | **4.44** | **3.02** |
| GLVQ-8D w/o companding (4-bit) | 5.75 | 5.14 | 4.12 | 3.54 | 5.06 | 4.48 | 3.06 |

# G   Ablation Study of Group Size

We vary the number of columns per group $\{32, 64, 128, 256, 512\}$ to study the trade-off between granularity and overhead (see Tables 9 and 10). Smaller groups (32, 64) offer slightly lower perplexity by adapting more precisely to local weight distributions, but they inflate model size and decoding cost because many lattice codebooks must be stored and reconstructed. Conversely, very large groups (256, 512) minimise codebook overhead yet fail to capture fine-grained distributional variation, yielding noticeably higher perplexity—particularly at 2- and 3-bit precision. A group size of **128** consistently provides the best balance, delivering the lowest (or near-lowest) perplexity while keeping storage and runtime overhead moderate; we therefore adopt 128 as the default in GLVQ.

Table 9: Ablation study of GLVQ with different group sizes on Wikitext2 using Llama 1-7B and 2-7B. We report perplexity ($\downarrow$) under different bit-widths.

| Group Size | Llama 1-7B (Wikitext2) | | | Llama 2-7B (Wikitext2) | | |
|---|---|---|---|---|---|---|
| | 2-bit | 3-bit | 4-bit | 2-bit | 3-bit | 4-bit |
| 32 | 6.26 | 5.71 | 5.68 | 5.65 | 5.10 | 4.97 |
| 64 | 6.26 | 5.72 | 5.70 | 5.66 | 5.13 | 5.00 |
| **128** | 6.28 | 5.72 | 5.70 | 5.69 | 5.13 | 5.01 |
| 256 | 6.34 | 5.78 | 5.75 | 5.84 | 5.19 | 5.08 |
| 512 | 6.47 | 5.85 | 5.82 | 5.91 | 5.28 | 5.17 |

Table 10: Ablation study of GLVQ with different group sizes on C4 using Llama 1-7B and 2-7B. We report perplexity ($\downarrow$) under different bit-widths.

| Group Size | Llama 1-7B (C4) | | | Llama 2-7B (C4) | | |
|---|---|---|---|---|---|---|
| | 2-bit | 3-bit | 4-bit | 2-bit | 3-bit | 4-bit |
| 32 | 7.80 | 7.16 | 7.04 | 7.30 | 6.38 | 6.28 |
| 64 | 7.81 | 7.17 | 7.06 | 7.32 | 6.39 | 6.31 |
| **128** | 7.83 | 7.17 | 7.08 | 7.33 | 6.40 | 6.31 |
| 256 | 7.95 | 7.25 | 7.15 | 7.42 | 6.50 | 6.39 |
| 512 | 8.14 | 7.32 | 7.18 | 7.51 | 6.66 | 6.54 |

# H   Ablation Study of Calibration Dataset Size

To evaluate data efficiency, we vary the calibration corpus from 100 K to 8 M RedPajama tokens (Appendix Table 11). Perplexity drops steadily until about 4 M tokens, after which gains saturate—an 8 M-token set brings no further improvement and occasionally introduces minor noise. Crucially, GLVQ remains robust with far fewer samples: reducing the set to 1 M or 500 K tokens incurs only a marginal increase in perplexity, and even a 100 K-token subset delivers competitive accuracy. These results confirm that GLVQ attains its best performance with roughly 4 M calibration tokens and degrades gracefully when calibration data are scarce, underlining its practicality in low-resource scenarios.

Table 11: Perplexity (↓) of GLVQ under different calibration set sizes, evaluated on Wikitext2 and C4 using Llama models (2-bit quantization, context length 2048). Results show that GLVQ achieves best performance at 4M tokens and remains stable across a wide range of calibration sizes.

| Calib Size | Tokens | Wikitext2 | | | | | | C4 | | | | | | |
|---|---|---|---|---|---|---|---|---|---|---|---|---|---|---|
| | | 1-7 | 1-13 | 1-30 | 1-65 | 2-7 | 2-13 | 1-7 | 1-13 | 1-30 | 1-65 | 2-7 | 2-13 | 2-70 |
| GLVQ-8D | 8M | 6.30 | 5.65 | 4.60 | 4.02 | 5.71 | 5.03 | 7.84 | 7.18 | 6.50 | 5.95 | 7.34 | 6.57 | 5.26 |
| **GLVQ-8D** | **4M** | **6.28** | **5.64** | **4.57** | **4.01** | **5.69** | **5.02** | **7.83** | **7.17** | **6.48** | **5.94** | **7.33** | **6.56** | **5.25** |
| GLVQ-8D | 2M | 6.34 | 5.69 | 4.63 | 4.06 | 5.74 | 5.07 | 7.87 | 7.23 | 6.54 | 6.00 | 7.38 | 6.61 | 5.30 |
| GLVQ-8D | 1M | 6.38 | 5.73 | 4.68 | 4.10 | 5.78 | 5.12 | 7.92 | 7.29 | 6.59 | 6.05 | 7.43 | 6.67 | 5.35 |
| GLVQ-8D | 500K | 6.45 | 5.80 | 4.74 | 4.16 | 5.84 | 5.19 | 8.00 | 7.36 | 6.67 | 6.11 | 7.50 | 6.74 | 5.42 |
| GLVQ-8D | 100K | 6.58 | 5.94 | 4.85 | 4.29 | 5.97 | 5.33 | 8.17 | 7.52 | 6.84 | 6.27 | 7.67 | 6.89 | 5.60 |

# I Babai Rounding vs. Greedy Coordinate Descent (GCD)

We replace Babai rounding with greedy coordinate descent (GCD) and re-train the quantizer (Appendix Tables 12 and 13). GCD exhibits unstable optimisation and poorer final perplexity across all model sizes: for 2-bit Llama 1-13B, perplexity rises from 5.64 (Babai) to 5.92 (GCD), and similar gaps persist at 3- and 4-bit settings. The results underline Babai rounding's advantage in both convergence speed and final quality, and we therefore retain it as the default index-assignment method in GLVQ.

Table 12: Comparison of GLVQ with Babai Rounding vs. GLVQ with Greedy Coordinate Descent (GCD) on Llama 1 & 2. Perplexity (↓) is reported on Wikitext2 and C4 for 4-, 3-, and 2-bit quantization.

| Method | Bits | Wikitext 2 | | | | | | C4 | | | | | |
|---|---|---|---|---|---|---|---|---|---|---|---|---|---|
| | | 1-7 | 1-13 | 1-30 | 1-65 | 2-7 | 2-13 | 1-7 | 1-13 | 1-30 | 1-65 | 2-7 | 2-13 |
| GLVQ-8D-babai | 4 | **5.70** | **5.08** | **4.09** | **3.52** | **5.01** | **4.44** | **7.08** | **6.55** | **5.92** | **5.58** | **6.31** | **5.81** |
| GLVQ-8D-GCD | 4 | 5.90 | 5.32 | 4.28 | 3.72 | 5.23 | 4.65 | 7.30 | 6.78 | 6.18 | 5.82 | 6.62 | 6.04 |
| GLVQ-8D-babai | 3 | **5.72** | **5.10** | **4.11** | **3.56** | **5.13** | **4.48** | **7.17** | **6.60** | **5.91** | **5.58** | **6.40** | **5.85** |
| GLVQ-8D-GCD | 3 | 5.95 | 5.37 | 4.36 | 3.82 | 5.38 | 4.72 | 7.41 | 6.83 | 6.13 | 5.78 | 6.70 | 6.08 |
| GLVQ-8D-babai | 2 | **6.28** | **5.64** | **4.57** | **4.01** | **5.69** | **5.02** | **7.83** | **7.17** | **6.48** | **5.94** | **7.33** | **6.56** |
| GLVQ-8D-GCD | 2 | 6.72 | 6.08 | 5.01 | 4.45 | 6.14 | 5.46 | 8.30 | 7.62 | 6.96 | 6.39 | 7.81 | 7.00 |

Table 13: Zeroshot Accuracy comparison of GLVQ with Babai Rounding vs. GLVQ with Greedy Coordinate Descent (GCD) on Llama 2.

| Method | 2-13 | | | | | 2-7 | | | | |
|---|---|---|---|---|---|---|---|---|---|---|
| | Bits | ArcC | ArcE | PIQA | WINO | Bits | ArcC | ArcE | PIQA | WINO |
| Original Model | 16 | 45.6 | 73.3 | 73.5 | 69.6 | 16 | 40.0 | 69.3 | 78.5 | 67.3 |
| GLVQ-8D-babai | 4 | **46.0** | **74.0** | **79.2** | **70.5** | 4 | **40.7** | **69.5** | **78.6** | **67.8** |
| GLVQ-8D-GCD | 4 | 45.3 | 73.4 | 78.4 | 69.8 | 4 | 39.9 | 68.9 | 77.8 | 67.0 |
| GLVQ-8D-babai | 3 | **46.5** | **74.2** | **79.3** | **71.0** | 3 | **40.9** | **69.8** | **78.8** | **67.9** |
| GLVQ-8D-GCD | 3 | 45.7 | 73.5 | 78.5 | 70.2 | 3 | 40.0 | 69.0 | 77.9 | 67.1 |
| GLVQ-8D-babai | 2 | **40.0** | **70.8** | **78.0** | **71.3** | 2 | **35.5** | **64.5** | **75.5** | **67.2** |
| GLVQ-8D-GCD | 2 | 38.9 | 69.4 | 76.7 | 70.0 | 2 | 34.0 | 63.2 | 73.9 | 66.0 |

