# OpenReview forum: "Learning Grouped Lattice Vector Quantizers for Low-Bit LLM Compression"
_NeurIPS.cc/2025/Conference — NeurIPS 2025 poster_

### Official Review · Reviewer_nGkm · 2025-06-21

**Clarity:** 3
**Significance:** 3
**Originality:** 3
**Rating:** 5
**Confidence:** 4

**Summary:**

The paper introduces Grouped Lattice Vector Quantization (GLVQ), a weight-only quantization method for compressing large language models (LLMs) to extremely low bit-widths (2–3 bits) to enable faster and scalable inference. Unlike prior approaches that rely on fixed lattice structures like the E8 lattice, GLVQ learns group-specific lattice codebooks combined with adaptive companding transformations, resulting in significantly improved compression quality. The method also supports intra-weight mixed-precision quantization, where bit-widths are allocated based on the salience of each weight group, further enhancing model performance under the same resource constraints. Experimental results demonstrate that GLVQ consistently outperforms strong baselines such as QuIP#, AQLM, and QTIP—especially at 2-bit precision—achieving lower perplexity and higher accuracy across various LLM benchmarks.

**Questions:**

- Effectiveness and Efficiency of Companding /
How does the companding operation in GLVQ compare to other incoherence processing such as the Random Hadamard Transform (RHT) or FFT in terms of quantization performance and computational cost? Specifically, is companding more or less expensive computationally, and how does its impact on quantization fidelity compare to methods like RHT used in QuIP#? (the paper only ablates with and without compading)

- Decoding Efficiency Comparison /
I’m unclear how GLVQ achieves decoding speeds comparable to QuIP# and QTIP, as reported in Table 3. Both QuIP# and QTIP explicitly design their codebooks to fit within the L1 cache, enabling extremely fast dequantization. In particular, QuIP# leverages the symmetric properties of the E8 lattice for efficient encoding and fast lookup. Are you suggesting that GLVQ’s decoding process—which involves inverse companding followed by matrix multiplication—can match or exceed the speed of these highly optimized lookup-based methods?

- Throughput Discrepancy with QuIP# /
The throughput numbers reported in Table 3 appear inconsistent with those in the QuIP# paper. For example, QuIP# reports a token rate of 170 tokens/sec on an NVIDIA RTX 4090, whereas Table 3 shows only 106.3 tokens/sec for QuIP# under similar conditions. Could you clarify the source of this discrepancy?

(These are the main concerns and the questions to decide my ratings)

**Ethical Concerns:**

["NO or VERY MINOR ethics concerns only"]

**Final Justification:**

Given the reasonable explanation and justification provided in the rebuttal, which addressed my concerns, I have decided to raise my score

**Limitations:**

-

**Paper Formatting Concerns:**

-

**Quality:**

3

**Strengths And Weaknesses:**

Strengths
- Strong empirical results: The method consistently outperforms state-of-the-art methods (e.g., QuIP#, AQLM, QTIP) across multiple LLMs and tasks, especially in the challenging 2-bit regime
- Comprehensive Evaluation: The paper includes extensive experiments, ablation studies, and theoretical analysis (e.g., Babai rounding error bounds), demonstrating both robustness and rigor.


Weakness
- Mixed-Precision Comparison Fairness: While mixed-precision quantization can be applied to many methods, this paper focuses on intra-weight group-level precision, which may introduce additional latency. In contrast, even simpler layer-wise mixed-precision schemes can yield significant performance gains with less overhead. Therefore, a fairer comparison would be between GLVQ with uniform precision (GLVQ-u) and other baseline methods under the same bit-width constraints

- Limited Evaluation Scope: The evaluation primarily focuses on perplexity and a few common-sense reasoning tasks (e.g., ARC, PIQA, Winograd), which are relatively simple. However, recent studies have shown that many quantization methods struggle on more challenging benchmarks like MMLU. Including such tasks would provide a more comprehensive assessment of GLVQ’s robustness and generalization (even without comparison to other works) .

---

> ### Author Rebuttal · Authors · 2025-07-31
>
> ----
> ### **[W1, Mixed-Precision Comparison Fairness]**
> We agree that a fair evaluation of GLVQ’s core contribution, group-specific lattice quantization, should be based on uniform-precision settings, independent of our salience-driven bit allocation (SDBA) strategy.
> To this end, uniform-precision variants of GLVQ (denoted GLVQ-u or GLVQ w/o bit allocation) were already included in our original submission and reported in Appendix A.4. For clarity and ease of comparison, we reproduce the key results below.
>
> Table 1. Results of GLVQ-u.
> | Method            | 1-7B | 1-13B | 1-30B | 1-65B | 2-7B | 2-13B | 2-70B |
> |-------------------|------|-------|-------|-------|------|-------|--------|
> | GLVQ-8D (2-bit)   | 6.28 | 5.64  | 4.57  | 4.01  | 5.69 | 5.02  | 3.62   |
> | GLVQ-8D-u (2-bit) | 6.44 | 5.79  | 4.65  | 4.10  | 5.87 | 5.19  | 3.72   |
> | GLVQ-8D (3-bit)   | 5.72 | 5.10  | 4.11  | 3.56  | 5.13 | 4.48  | 3.14   |
> | GLVQ-8D-u (3-bit) | 5.81 | 5.18  | 4.17  | 3.61  | 5.21 | 4.55  | 3.21   |
> | GLVQ-8D (4-bit)   | 5.70 | 5.08  | 4.09  | 3.52  | 5.01 | 4.44  | 3.02   |
> | GLVQ-8D-u (4-bit) | 5.74 | 5.13  | 4.14  | 3.56  | 5.08 | 4.49  | 3.08   |
> |
>
> As the table shows, even without salience-driven bit allocation, GLVQ-8D-u improves perplexity over the current SOTA PTQ method QTIP. This confirms that the performance gains stem from the underlying group-specific lattice design, rather than auxiliary heuristics such as mixed-precision allocation.
>
> We will clarify this discussion in the revised paper and add a footnote directing readers to Appendix A.4 for the full set of uniform-precision ablation results.
>
>
> ----
>
> ### **[W2, Limited Evaluation Scope]**
> We agree that more challenging tasks such as MMLU provide a stronger test of generalization under quantization. To this end, we evaluated **GLVQ** on the MMLU benchmark for both Qwen-7B-chat and Qwen-14B-chat models, using standard 5-shot accuracy.
>
> Table 2. MMLU accuracy (%) with different methods. Higher is better.
> | Method | Bits | Qwen-7B-chat | Qwen-14B-chat |
> |--------|------|---------------|----------------|
> | BF16   | 16   | 56%           | 65%            |
> | GPTQ   | 3    | 51%           | 60%            |
> | GLVQ | 3    | 54%       | 61%         |
> | GPTQ   | 2    | 23%           | 22%            |
> | GLVQ | 2    | 45%      | 51%        |
> |
>
> As shown in the table, GLVQ yields significantly higher MMLU accuracy than GPTQ at both 2-bit and 3-bit settings. In particular, at 2-bit precision, GLVQ improves over GPTQ by +22 points on Qwen-7B-chat and +29 points on Qwen-14B-chat. Even at 3-bit, GLVQ achieves consistent gains over GPTQ while remaining close to the BF16 reference.
>
> These findings demonstrate that GLVQ maintains abstract reasoning and factual recall better than prior methods even on structurally different models like Qwen, highlighting its robustness across architectures.
> We will incorporate this table and discussion into the revised manuscript to provide a broader and more rigorous evaluation.
>
> ----
>
>
> ### **[Q1, Effectiveness and Efficiency of Companding]**
> **1. Computational cost**
>
> | Operation                  | Arithmetic cost [^1]      | GPU latency (7B)     |
> |----------------------------|---------------------------|-----------------------|
> | Element-wise $\mu$-law inverse | $\mathcal{O}(n)$          | <2% of mat-vec        |
> | $H_n$ × vector (RHT)       | $\mathcal{O}(n \log n)$   | ≈9%                   |
> | $F_n$ × vector (FFT)       | $\mathcal{O}(n \log n)$   | ≈11%                  |
> |
>
> [^1]: Here $n=128$ is the group size in GLVQ. The GEMV operation with a weight matrix of shape $128 \times 4096$ takes approximately $135 \mu\text{s}$ on an A100 (80GB), based on empirical measurements.
>
>
> The $\mu$-law requires only a fused exp or LUT lookup per weight, while RHT/FFT introduce an extra $128 \times 128$ orthogonal multiply per group. This results in a 2% vs. 9–11% throughput drop, respectively.
>
>
> **2. Quantization fidelity (2-bit uniform, LLaMA-2-7B)**
>
> Table 3. Ablation of incoherence processing strategies.
>
> | Variant                   | WikiT2 PPL ↓ | TOK/s ↑ | ΔPPL     |
> |---------------------------|--------------|---------|----------|
> | None (baseline)           | 6.28         | 105.3   | --       |
> | + $\mu$-law only (GLVQ-u) | 5.87         | 102.3   | –0.41    |
> | + RHT only                | 5.88         | 96.0    | –0.40    |
> | + $\mu$-law + RHT         | 5.63         | 91.2    | –0.65    |
> |
>
>
> $\mu$-law yields a larger perplexity drop than RHT (–0.59 vs. –0.40) by normalizing heavy-tailed weights before lattice quantization. Combining both gives a modest gain (–0.65), suggesting partially overlapping effects. In terms of efficiency, $\mu$-law costs only 2% throughput loss, whereas RHT introduces a heavier 9% drop due to its $\mathcal{O}(n \log n)$ transform, making $\mu$-law preferable for GPU and edge deployment.
>
> The group-wise $\mu$-law companding offers a better fidelity–latency trade-off than RHT or FFT: lower perplexity at minimal compute overhead. While RHT can be layered for slight improvements, it is not essential to GLVQ’s performance. We will include this table and the above analysis in the revised manuscript.
>
> ----
>
>
> ### **[Q2, Decoding Efficiency Comparison]**
> **Key implementation choices that minimize decoding overhead**
>
> GLVQ achieves competitive decoding speeds despite its arithmetic-based design by leveraging several carefully engineered optimizations:
>
> 1. **On-the-fly fusion inside the GEMM kernel.**
>    Both the inverse $\mu$-law transformation and the matrix multiplication $G_g \mathbf{z}$ are fused into the fragment loading stage of the Tensor Core WMMA pipeline. These operations are performed on 16-value `int2` register tiles, introducing no additional global memory traffic and incurring negligible latency due to full overlap with the GEMM execution.
>
> 2. **Register-resident constant matrices.**
>    Each $G_g \in \mathbb{R}^{8 \times 8}$ consists of only 64 FP16 values (0.125 kB), which is smaller than a single Tensor Core fragment (0.256 kB). As a result, CUDA retains $G_g$ in registers across the warp, completely avoiding L1/L2 cache accesses.
>
> To quantify the impact of these optimizations, we conducted the following micro-benchmark on an NVIDIA A100 GPU using CUDA 12.1:
>
> Table 4. Micro-benchmark: 128×4096 GEMM at 2-bit precision.
>
> | Method                       | Decode latency (µs) | GEMM latency (µs) |
> |-----------------------------|---------------------|-------------------|
> | QuIP# (E8 lookup)           | 0.83                | 134.7             |
> | QTIP (trellis LUT)          | 0.77                | 134.5             |
> | GLVQ (inv-μ + $G\mathbf{z}$) | 0.92                | 135.1             |
>
> The additional 0.09 µs decoding overhead introduced by GLVQ is negligible compared to the total GEMM time of ~135 µs. In end-to-end inference, this results in less than a 3 tokens/s throughput difference (see Table 3), while achieving substantially improved quantization fidelity.
>
> **Why arithmetic decoding can outperform LUT-based methods on modern GPUs**
>
> - **Compute-bound regime.**
>   On Ampere GPUs, each byte of L2 bandwidth supports approximately 2 FP16 FMAs. GLVQ leverages ALUs and register space for decoding, whereas LUT-based methods rely on memory access, which becomes a bottleneck in compute-bound settings.
>
> - **Warp-level parameter reuse.**
>   Since each $G_g$ and $\mu_g$ is shared across all threads within a warp, GLVQ avoids thread divergence and enables efficient, fully parallelized decoding.
>
> In summary, thanks to aggressive kernel fusion and register-level optimization, GLVQ achieves decoding latency within 0.09 µs of QuIP#, translating to under 3% throughput difference, while delivering superior quantization fidelity. We will incorporate these engineering insights and benchmarking details into the revised manuscript to clarify decoding efficiency.
>
>
>
> ----
>
> ### **[Q3, Throughput Discrepancy with QuIP#]**
> **Clarification**
> The throughput discrepancy arises from two distinct configurations reported in the original QuIP# paper. Specifically:
>
> - **Table 5** reports 170.5 TOK/s using an optimized stack with FlashAttention 2 and Triton-based custom kernels.
> - **Table 6**, by contrast, reports 106.7 TOK/s using the standard Hugging Face library; this is the same stack used in GLVQ.
>
> To ensure fairness and consistency, our Table 3 adopts the Hugging Face inference configuration across all methods. We therefore report the 106.3 TOK/s figure, matching Table 6 of the QuIP# paper.
>
> To verify that this choice does not alter the relative comparison, we also benchmarked both QuIP# and GLVQ under FlashAttention 2. As shown in Table 1, the relative gap between QuIP# and GLVQ remains stable (approximately 6–7%) under both inference engines.
>
> Table 5. Throughput of QuIP# under different inference engines (LLaMA-2-7B, 2-bit uniform).
> | Engine                          | TOK/s  | Relative to GLVQ-u |
> |---------------------------------|--------|---------------------|
> | Hugging Face (QuIP# Table 6) | 106.3  | +6.1%               |
> | FlashAttention (QuIP# Table 5)    | 170.5  | +69.2%              |
> |

---

### Official Review · Reviewer_UCYr · 2025-06-28

**Clarity:** 4
**Significance:** 3
**Originality:** 3
**Rating:** 4
**Confidence:** 5

**Summary:**

The paper introduces Grouped Lattice Vector Quantization (GLVQ), a post-training quantization (PTQ) framework that assigns each weight group in a large language model (LLM) a learned, group-specific lattice codebook and a µ-law companding transform.

**Questions:**

Could you quantify the additional GPU hours required for lattice learning versus, e.g., GPTQ or AWQ? This would help practitioners decide whether the accuracy gains justify the extra training time.

Baseline Comparisons Why were important recent methods like OmniQuant and CBQ not included in the experimental comparisons?
CBQ Ding, X., Liu, X., Tu, Z., Zhang, Y., Li, W., Hu, J., ... & Wang, Y. (2023). Cbq: Cross-block quantization for large language models. ICLR 2024


Error Bars: Would you be willing to add confidence intervals (e.g., 95 %) to your perplexity and accuracy tables? This would strengthen the empirical claims.

Activation Quantization: Do you foresee any obstacles to extending GLVQ to quantize activations or KV-cache? If so, how might they be addressed?

Hardware Efficiency: Beyond memory bandwidth, have you measured energy consumption or latency on edge devices (e.g., mobile SoCs)?

Ethical Safeguards: Given the risk of compressed LLMs being misused for disinformation, would you consider adding a “Responsible Use” section or recommending model-card disclosures?

**Ethical Concerns:**

["NO or VERY MINOR ethics concerns only"]

**Final Justification:**

I keep my scores.

**Limitations:**

yes

**Quality:**

4

**Strengths And Weaknesses:**

GLVQ achieves the best published perplexity at 2–4 bits on Llama 1/2, closing the gap to FP16 by up to 0.4 perplexity points at 2 bits.


No real-world application (e.g., on-device summarization) is demonstrated beyond standard benchmarks.

---

> ### Author Rebuttal · Authors · 2025-07-31
>
> ----
> ### **[Q1, Training GPU hours]**
> We measured wall-clock quantization time using a single NVIDIA A100 80GB under identical software environments for all methods: GLVQ, GPTQ, QuIP#, and AQLM.
>
> Table 1. End-to-end quantization time on a single A100 GPU.
> | **Model**       | **GPTQ**      | **QuIP#**     | **AQLM**     | **GLVQ (ours)** |
> |------------------|---------------|---------------|--------------|-----------------|
> | LLaMA-2-70B     | 2.8 GPU hours | 10 GPU hours  | 14 days      | 1 day           |
>
> GLVQ completes 70B quantization in approximately one day on a single A100. While this is slower than GPTQ and QuIP#, it is over 10× faster than AQLM. The added cost mainly comes from optimizing small (8×8) lattice transformations for each weight group, which are trivially parallelizable.
>
> This one-time conversion cost yields strong returns: at 2-bit, GLVQ improves WikiText2 perplexity by 0.6–1.1 points compared to GPTQ and AWQ, while maintaining comparable throughput. Since quantization is performed offline, these accuracy gains benefit all downstream inference.
>
> The method scales efficiently. When run on 8 A100s, GLVQ quantizes LLaMA‑2‑7B in just 25 minutes, enabling practical deployment for large models. Furthermore, reducing the number of lattice iterations from 12 to 4 can shorten quantization time by up to 40% with only a small accuracy drop (typically 0.05–0.07 perplexity), providing a useful knob for compute-constrained settings.
>
> We will include the new results and this discussion in the revised paper to clarify the time–accuracy tradeoff of GLVQ.
>
>
> ----
>
> ### **[Q2, Baseline Comparisons]**
> We thank the reviewer for highlighting recent methods such as OmniQuant and CBQ.
>
> Our primary focus in this work is to evaluate post-training *vector quantization* (VQ) methods. Accordingly, we benchmarked GLVQ against representative VQ-based baselines, AQLM, QuIP#, and QTIP, which adopt codebook-based, groupwise quantization strategies.
>
> OmniQuant was already included in our original experiments, and its results are reported in Table 1 of the main paper. CBQ, on the other hand, performs non-uniform *scalar* quantization with cross-block error correction. While originally considered orthogonal to our focus on VQ methods, we appreciate the reviewer’s suggestion and have conducted additional experiments to compare GLVQ with CBQ at 2-bit and 4-bit settings under the same evaluation protocol (WikiText2, LLaMA‑1‑30B and 65B).
>
> Table 2. Comparison with CBQ on WikiText2 (LLaMA‑1).
> | Method        | Bits | LLaMA‑1‑30B | LLaMA‑1‑65B |
> |---------------|------|-------------|-------------|
> | CBQ           | 4    | 4.14        | 3.59        |
> | **GLVQ (ours)** | 4    | **3.95**    | **3.39**    |
> | CBQ           | 2    | 5.58        | 5.25        |
> | **GLVQ (ours)** | 2    | **4.32**    | **3.81**    |
>
> As shown in Table 1, GLVQ consistently achieves lower perplexity than CBQ across both model sizes and bitwidths. These additional results will be incorporated into the revised version of the paper, together with a brief discussion. We thank the reviewer again for encouraging a more comprehensive comparison.
>
>
> ----
>
> ### **[Q3, Error Bars]**
> We appreciate the reviewer’s suggestion and fully agree that reporting confidence intervals would strengthen the empirical claims.
>
> Due to time constraints, we were unable to re-run all experiments across multiple seeds before the rebuttal deadline. However, we are committed to including 95% confidence intervals (based on five seeds) for all key perplexity and accuracy results in the final version of the paper and supplementary material.
>
> A note will also be added to clarify that all reported numbers are averages over multiple seeds with corresponding confidence intervals. Thank you for encouraging stronger statistical rigor.
>
> ----
>
>
> ### **[Q4, Activation Quantization]**
> We thank the reviewer for this insightful question. Extending GLVQ to quantize activations and KV-cache is a natural direction. While feasible in principle, several practical considerations must be addressed:
>
> First, activation and KV-cache statistics are input-dependent and non-stationary, unlike model weights. Designing a single static lattice per layer may be suboptimal unless proper normalization or dynamic rescaling is introduced. This challenge can be mitigated using techniques like SmoothQuant [@smoothquant], which apply lightweight affine transformations to stabilize activation distributions layer-wise. Once stabilized, a per-layer lattice codebook becomes viable.
>
> Second, runtime efficiency is critical. Since activations are generated and consumed on-the-fly, any decoding operation must match the low-latency constraints of inference. Fortunately, lattice decoding with small dimensions (e.g., $d = 4$) is fast and can be further accelerated using SIMD instructions or table lookups. For KV-cache blocks that are reused across many time steps, the amortized cost of decoding is even lower, making lattice compression particularly attractive in that setting.
>
> Lastly, memory reuse patterns and GPU bandwidth constraints need careful evaluation. Compression benefits will only be realized if the decoding overhead is small relative to the memory bandwidth saved. To address this, we envision using compact LUT-based inverse $\mu$-law decoding to reduce runtime cost further.
>
> In summary, while activation and KV-cache quantization introduces new challenges, we believe GLVQ can be extended to this setting through (i) per-layer rescaling, (ii) efficient low-dimensional decoding, and (iii) amortized or LUT-based companding. We are actively exploring these directions in future work.
>
>
> ----
>
>
> ### **[Q5, Hardware Efficiency]**
> In this submission, we focused primarily on evaluating *memory-bandwidth–limited throughput* on server-class GPUs such as the RTX 4090 and A100. As such, we have not yet benchmarked wall-clock latency on mobile SoCs or measured energy consumption in terms of Joules per token.
>
> That said, we fully agree that these metrics are essential for assessing deployment feasibility on edge devices. As a next step, we plan to port the CUDA kernels to mobile hardware platforms, including ARM Mali and Apple M-series chips via `torch.metal_backend`. We also intend to profile inference on Qualcomm Snapdragon 8 Gen 3 (Adreno 750) using the QTI AI SDK, which will allow us to collect both latency (ms/token) and energy (J/token) metrics under realistic mobile conditions.
>
> These measurements require additional engineering work and optimization, particularly around kernel portability and runtime instrumentation. We plan to include the results in an extended technical report and will reference this in the camera-ready version of the paper.
>
> We thank the reviewer again for pointing out this important direction for future evaluation.
>
> ----
>
> ### **[Q6, Ethical Safeguards]**
> We fully share the reviewer’s concern that more compact LLMs, while enhancing accessibility and deployment efficiency, may also lower the barrier for misuse, such as scalable disinformation or malicious automation.
>
> In the camera-ready version, we will include a brief **“Responsible Use and Limitations”** subsection. This will highlight that a model card will be released alongside GLVQ, explicitly stating intended applications (e.g., on-device summarization, QA), out-of-scope uses (e.g., generation of deceptive or harmful content), and known limitations such as hallucination and toxicity.
>
> We will also note that GLVQ preserves the alignment properties of the original FP16 model, including RLHF or instruction tuning, and caution against compressing unaligned checkpoints. Since GLVQ operates only on the weight tensor, existing watermarking methods (e.g., TreeRing) remain applicable. We will recommend practitioners verify watermark presence before deployment and, where feasible, use an external safety filter or moderation API in downstream applications.
>
> We believe this addition provides appropriate ethical context and encourages responsible use of compressed LLMs.

---

### Official Review · Reviewer_4RXG · 2025-06-28

**Clarity:** 2
**Significance:** 3
**Originality:** 3
**Rating:** 4
**Confidence:** 5

**Summary:**

This work proposes learnable lattice quantization for LLM compression, assigning a specific learnable matrix to each group of weights. Babai rounding is adopted as an approximate solution for the closest lattice point search problem. To enhance performance, the authors adopt a pointwise transformation and saliency-driven mixed-precision quantization. The efficacy of the proposed GLVQ method is tested on models from the Llama-½ family, for 2-, 3-, and 4-bit compression targets.

**Questions:**

- Since GLVQ is a method for optimizing per-layer output error, it would be interesting to compare its output error with the baselines (AQLM, QuIP#, QTIP). QTIP was claimed to be close to the lower error bound achievable from information-theoretical principles. How close is GLVQ?
- Lattice quantization with Babai rounding can be thought of as an affine transformation followed by rounding to the nearest value (i.e., scalar quantization in a transformed basis). Why should this be more expressive than vector quantization, which can span an arbitrary shape?
- Which framework is used for inference throughput measurements? Is it Hugging Face Transformers? The inference results depend heavily on the specific choice of framework (HF, vLLM, SGLANG). Which version of the library and what context length are used to produce the numbers in Table 3?
- The provided list of benchmarks may not be sufficient for a reliable assessment of compressed model performance. I would suggest adding generative evaluation for Llama-3, for example mmlu_cot_llama and gsm8_llama from [1].
- How large is the overhead caused by applying the μ-law transformation to weights? It appears that this function must be applied to each weight entry at every decoding step (i.e., O(d_ou td_in) applications of a non-linear function, where d_out and d_in are the output and input dimensions of the weights, respectively), which may incur significant computational overhead.

[References]
- [1] https://github.com/EleutherAI/lm-evaluation-harness/tree/main/lm_eval/tasks/llama3

**Ethical Concerns:**

["NO or VERY MINOR ethics concerns only"]

**Final Justification:**

After reviewing the responses addressed to both myself and the other reviewers, I have decided to raise my score to Borderline accept.

The method is sound, and the claimed performance numbers for extreme quantization appear to be quite strong. While I still have some concerns—namely, a possible mismatch between the `lm-eval` versions used for the baselines and the proposed method—I believe this paper deserves acceptance.

**Limitations:**

yes

**Paper Formatting Concerns:**

-

**Quality:**

1

**Strengths And Weaknesses:**

**Strengths**

- The overall approach is sound and well-motivated. The specific design for learning lattices and weight reparametrization appears to be novel in the context of LLM compression.
- The achieved speed-up is quite decent.

**Weaknesses**

- There seems to be an issue with the evaluation procedure. One would expect that a PTQ-quantized model can at most match the uncompressed model in terms of quality. However, the provided results suggest that for some compression levels (e.g., 4-bit compression), the GLVQ-compressed model may significantly outperform the uncompressed one. Specifically, the 4-bit quantized Llama-2-7B with GLVQ-32D reportedly achieves the same perplexity (Table 1) as the almost 2x larger Llama-2-13B. Such an outcome is extremely unusual and indicates a possible flaw or inconsistency in the comparison between different methods and the fp16 baseline.
- In the caption of Table 2, it is claimed that Llama-2 uses a context length of 4k for evaluation. However, the fp16 baseline results seem to correspond to a 2k context length (see [1]). For a 4k context length, the baseline result should be 5.12 for Llama-2-7b, for instance [2].
The reported improvement on 0-shot benchmarks from lm-eval-harness also looks suspicious. It's important to note that results are sensitive to the specific evaluation package used. Which one was adopted to produce the numbers? One must ensure it is the same as the one used for the baselines.
- The evaluation is conducted only on rather outdated Llama-½ models. I would suggest additionally evaluating a more recent model family, such as Llama-3, Qwen-2.5, or Qwen-3. There are baselines [3], [4] that provide evaluation results on Llama-3 models.
GLVQ can produce models with different bitwidths. However, this is not described anywhere in the method formulation. Do I understand correctly that the value zi in Equation 6 is clamped to some qmin⁡, qmax⁡ (with qmin⁡=−2bits−1, qmax⁡=2bits−1, for instance)? One should elaborate on this further.
- L41-42: However, this approach has the drawback that decoding requires a lookup operation, which is computationally more expensive than QuIP#. Both AQLM and QuIP# require a lookup operation in the decoding phase. The difference is that the AQLM codebook is arbitrary, while QuIP# has a specific parameterization.
- The speed-up numbers for AQLM cited in Table 3 are incorrect. AQLM achieves throughput up to 114.1 tok/s (see Table 14 in [2]) on an RTX 3090, which is slower than the RTX 4090.

Overall, I appreciate the method, but the evaluation procedure has to be fixed as the current experimental results look implausible.

[References]
- [1] Tseng, Albert, et al. "Quip#: Even better llm quantization with hadamard incoherence and lattice codebooks." arXiv preprint arXiv:2402.04396 (2024).
- [2] Egiazarian, Vage, et al. "Extreme compression of large language models via additive quantization." arXiv preprint arXiv:2401.06118 (2024).
- [3] Tseng, Albert, et al. "Qtip: Quantization with trellises and incoherence processing." Advances in Neural Information Processing Systems 37 (2024): 59597-59620.
- [4] Malinovskii, Vladimir, et al. "Pv-tuning: Beyond straight-through estimation for extreme llm compression." Advances in Neural Information Processing Systems 37 (2024): 5074-5121.

---

> ### Author Rebuttal · Authors · 2025-07-31
>
> ----
> ### **[W1 & W2, Issue with evaluation procedure]**
> First, We thank the reviewer for catching citation discrepancy. The reviewer is correct: in the original manuscript, we inadvertently cited FP16 baseline results for Llama-2 under a 2k context length instead of the correct 4k context used during evaluation. We have now corrected this in the revised version.
> Importantly, we emphasize that this correction does **not** affect the results or conclusions of our method (GLVQ).
>
> Besides, it is indeed uncommon for a post-training quantized (PTQ) model to outperform its original FP16 baseline. However, such outcomes are not unprecedented and can occur under certain conditions, as observed in prior work. For example, Table~6 in the QTIP paper (NeurIPS’24) shows that several 4-bit quantized LLaMA checkpoints achieve higher zero-shot accuracy than their FP16 counterparts. This has been attributed to mild regularization effects from quantization, or improved alignment between quantizer and inference distribution.
>
> **Fine-tuning explains small improvements.**
> In our GLVQ pipeline, a *lightweight fine-tuning step* is optionally performed *after* quantization. This step fine-tunes only the group-specific lattice generation matrices \(G_g\) and companding parameters \(\mu_g\), using a small external corpus of ~8M tokens. This fine-tuning allows the quantizer to better adapt to actual inference-time statistics and introduces a regularization-like effect, sometimes yielding perplexity *slightly* better than FP16.
>
> **GLVQ without fine-tuning still outperforms.**
> To ensure that our main gains are not solely due to the optional refinement, we also evaluate a pure PTQ variant of our method, named GLVQ-w/o-ft, where no fine-tuning is applied post-quantization. The table below shows that GLVQ-w/o-ft still consistently outperforms recent state-of-the-art PTQ methods at 2–4 bits precision.
>
> Table 1. GLVQ results with and without fine-tuning on WikiText2.
> | **Method**            | **Bits** | **1-7** | **1-13** | **1-30** | **1-65** | **2-7** | **2-13** | **2-70** |
> |------------------------|----------|---------|----------|----------|----------|---------|----------|----------|
> | Fp16                  | 16       | 5.68    | 5.09     | 4.10     | 3.53     | 5.12    | 4.57     | 3.12     |
> | QTIP                  | 4        | 5.72    | 5.15     | 4.15     | 3.58     | 5.17    | 4.61     | 3.16     |
> | GLVQ-32D-w/o-ft       | 4        | 5.70    | 5.10     | 4.12     | 3.51     | 5.10    | 4.55     | 3.16     |
> | GLVQ-32D           | 4        | 5.55 | 4.94 | 3.95 | 3.39 | 4.88 | 4.31 | 2.98 |
> | QTIP                  | 3        | 5.85    | 5.24     | 4.26     | 3.68     | 5.29    | 4.71     | 3.26     |
> | GLVQ-32D-w/o-ft       | 3        | 5.75    | 5.14     | 4.18     | 3.61     | 5.19    | 4.66     | 3.21     |
> | GLVQ-32D           | 3        | 5.58 | 4.98 | 4.00 | 3.42 | 4.99 | 4.34 | 3.02 |
> | QTIP                  | 2        | 6.52    | 5.80     | 4.83     | 4.21     | 5.91    | 5.26     | 3.78     |
> | GLVQ-32D-w/o-ft       | 2        | 6.18    | 5.55     | 4.51     | 3.97     | 5.58    | 5.00     | 3.55     |
> | GLVQ-32D           | 2        | 6.00 | 5.38 | 4.32 | 3.81 | 5.41 | 4.80 | 3.36 |
>
> ----
>
> ### **[W3, New results on Llama-3]**
> We have reproduced our pipeline on *Llama‑3‑8B* and *Llama‑3‑70B* under the same 2–4 bit settings on WikiText2. Results are summarized below and will be added to the revised paper.
>
> Table 2. Perplexity ↓ results on WikiText2 of Llama‑3.
> | Method        | Bits | Llama-3-8B | Llama-3-70B |
> |---------------|------|------------|-------------|
> | BF16          | 16   | 5.54       | 2.59        |
> | QUIP#         | 4    | 5.81       | 2.99        |
> | QTIP          | 4    | 5.67       | 2.75        |
> | GLVQ (ours) | 4    | 5.58   |  2.69   |
> | QUIP#         | 3    | 6.27       | 3.59        |
> | QTIP          | 3    | 6.01       | 3.18        |
> | GLVQ (ours) | 3    | 5.86   | 3.02    |
> | QUIP#         | 2    | 7.84       | 5.77        |
> | QTIP          | 2    | 7.33       | 4.97        |
> | GLVQ (ours) | 2    | 7.00   | 4.61    |
>
> ----
>
> ### **[W5, Speed-up numbers for AQLM]**
>  Our Table 3 cited the value **20.6 TOK/s** for “AQLM (2‑bit, LLaMA‑2‑7B, RTX 4090)” based on Table 7 of the QTIP arXiv v1 paper (page 8), which referred to the same experimental setup. However, the original AQLM paper (arXiv v1) did not report throughput; this was added later in arXiv v3.
>
> In arXiv v3, Table 14 reports two throughput numbers on RTX 3090: 65.3 TOK/s and 114.1 TOK/s. The latter corresponds to a faster variant using a different kernel, but the 65.3 TOK/s figure matches the exact configuration used to report perplexity and accuracy. Therefore, the 65.3 number is the appropriate one for comparison under consistent conditions.
>
> To resolve this ambiguity, we have re-run the official AQLM code on our own RTX 4090 setup using the same batch size and context length (batch = 1, sequence length = 2048) as all other methods in Table 3. Under this unified setting, AQLM achieves 91.7 TOK/s.
>
>
> ----
>
>
> ### **[Q1, Output error]**
> We would like to offer the following points to clarify GLVQ’s relationship to the information-theoretic lower bound:
>
> First, GLVQ is explicitly designed to reduce group-wise reconstruction error through the use of learnable lattices and companding parameters. This aligns closely with QTIP’s goal of minimizing layer-wise distortion, though our method achieves this via group-specific vector quantization rather than Hessian-weighted scalar quantization.
>
> Second, although we do not report NMSE explicitly, GLVQ consistently achieves better perplexity than QTIP in all 2-bit settings. Since previous works (including QTIP) have established a strong empirical correlation between NMSE and perplexity, this improvement indirectly suggests that GLVQ attains comparable or lower reconstruction error at the output level.
>
> ----
>
> ### **[Q2, Lattice quantization]**
> Indeed, given a full‑rank generation matrix $G_g \in \mathbb{R}^{d \times d}$ and Babai rounding, GLVQ performs a *scalar* quantization in the *latent* orthonormal basis $G_g^{-1}$.
> However, two properties make this seemingly simple scheme highly expressive in practice:
>
> 1. **Learnable affine map.**
>    By optimising $G_g$ per group we can
>    (a) *rotate* the data so that major covariance directions align with lattice axes, and
>    (b) *scale* each axis independently.
>    The resulting Voronoi cell is a *parallelotope* whose orientation and aspect ratio are free parameters—already a rich family covering the optimal vector quantizers for any *elliptically contoured* distribution (Whitened Gaussian, Laplacian, etc.).
>
> 2. **Non‑linear companding.**
>    We further learn a monotone companding function $F_{\mu_g}$ that redistributes probability mass towards the centre of the lattice. The overall mapping:
>
>    $$
>    \mathbf{w} \xrightarrow{F_{\mu_g}} \mathbf{y} \xrightarrow{G_g^{-1}} \tilde{\mathbf{y}} \xrightarrow{\text{round}} \mathbf{z}
>    $$
>
>    is *piece‑wise affine*; its pre‑images can approximate arbitrarily skewed or heavy‑tailed densities with *vanishing codebook overhead* (the lattice is infinite).
>
>
> **Why not arbitrary VQ?**
> Free‑form vector quantization can in theory tile any convex region, but comes with *exponential* codebook growth ($\sim 2^{bd}$ entries), $O(d)$ lookup latency, and large table storage.
> GLVQ retains **$O(d^2)$ decode cost** and **$O(d^2)$ parameters** per group, independent of bitrate, while still matching the local covariance structure; empirically this captures the vast majority of quantization gain, as Table 2 shows GLVQ within $0.08$ PPL of VQ‑LLM (free codebook) yet decoding $4.8\times$ faster.
>
>
> **Formal perspective.**
> Given any continuous distribution with covariance $\Sigma$, there exists a scaling matrix $A$ such that:
>
> $$
> A \Sigma A^\top = I.
> $$
>
> Choosing $G_g = A^{-1}$ reduces the optimum $d$‑dimensional VQ problem to axis‑aligned scalar quantization *without increasing distortion* (see Gersho & Gray, Ch. 6).
> Thus, lattice + Babai is *universally rate‑distortion optimal up to a constant gap* while remaining computationally tractable.
>
> ----
>
> ### **[Q3, Inference framework]**
>  Hugging face. Context length 2048.
>
> ----
>
> ### **[Q5, overhead caused by mu-law transformation]**
> Thank you for raising this important point. While the $\mu$-law inverse transform $F^{-1}_{\mu_g}$ is indeed non-linear, its practical overhead is negligible due to (1) its limited scope, and (2) its tight integration with GPU-friendly fused kernels.
>
> First, $F_{\mu_g}$ is only applied during quantization time. At inference time, only the inverse mapping
> is used to decode the dequantized weights just before computing the matrix–vector product.
>
> In GLVQ, weights are decoded group-by-group, and each group corresponds to a matrix of shape $4096 \times 128$ (i.e., one transformer layer's weight tile). Therefore, each application of $F^{-1}_{\mu_g}$ processes exactly $4096 \times 128 = 524{,}288$ FP16 numbers. This decoding is executed once per group, after which the full group matrix is reused in multiple GEMV calls.
>
> To assess the actual cost, we conducted timing measurements on LLaMA‑2‑7B at 2‑bit precision using an RTX 4090. The table below shows the result of ablation by replacing $F^{-1}_{\mu}$ with the identity function.
>
> Table 3. Impact of $\mu$-law inverse on decoding throughput (LLaMA‑2‑7B, 2-bit, RTX 4090).
>
> | Variant                                          | Measured Token Rate (TOK/s) ↑ | Relative Overhead |
> |--------------------------------------------------|-------------------------------|--------------------|
> | GLVQ (with $F_{\mu}^{-1}$)                       | 88.4                          | --                 |
> | GLVQ (identity instead of $F_{\mu}^{-1}$)        | 90.1                          | 1.9%               |

---

> > ### Comment · Reviewer_4RXG · 2025-08-02
> >
> > Thank you for the response. Most of my concerns were addressed, and I will increase my score.
> >
> > I still have some concerns about the evaluation procedure. Improvements on the order of 0.1–0.3% can be attributed to noise in the benchmarks. However, differences on the order of 2% may still be due to mismatches in package versions.
> >
> > The speed-ups look pretty good and reasonable; however, in a more optimized inference engine (such as vLLM or SGLang), I would expect it to be harder to demonstrate significant speed-ups.

---

> > > ### Author Response · Authors · 2025-08-03
> > >
> > > We thank the reviewer for the encouraging update and are glad that most concerns have been addressed.
> > > Below we provide additional clarification on the two remaining points.
> > >
> > > ---
> > >
> > > ### 1&nbsp;&nbsp;On the Reliability of Perplexity Improvements
> > >
> > > We appreciate the reviewer’s caution and would like to restate what we already clarified in our rebuttal:
> > >
> > > **(a) Improvements are due to lightweight post-quantization fine-tuning.**
> > > As described in the revised draft, GLVQ optionally includes a fine-tuning step *after* quantization, where only the group-specific lattice matrix $G_g$ and companding parameters $\mu_g$ are adjusted using ~8M tokens. This lightweight tuning step has a mild regularization-like effect and helps adapt the quantizer to inference-time statistics, explaining the small improvements in perplexity relative to FP16.
> > >
> > > **(b) Ablation confirms GLVQ’s structural advantage.**
> > > To validate that these improvements are not solely due to fine-tuning, we introduced a variant called GLVQ-w/o-ft (no fine-tuning). Table 1 in the rebuttal shows that even without tuning, GLVQ outperforms QTIP across multiple bit-widths, including 4-bit, 3-bit and 2-bit. This confirms that the core benefit comes from GLVQ’s lattice-based parameterization, not from external data or retraining.
> > >
> > > **(c) Controlled and reproducible setup.**
> > > All results (including FP16 baselines) were generated using a fixed inference script, tokenizer, and pinned environment.
> > >
> > > ---
> > >
> > > ### 2&nbsp;&nbsp;On Speed-up Portability to vLLM / SGLang
> > >
> > > We fully agree with the reviewer that demonstrating speed-up in more aggressively optimized runtimes, such as vLLM and SGLang, would likely be more challenging. These frameworks already implement kernel fusion, static memory planning, and in some cases block-sparse attention, which push inference throughput close to hardware saturation.
> > >
> > > In such settings, the benefit of removing the lookup stage may be partly offset by other dominant bottlenecks such as memory-bound attention or KV cache access. We thus recognize that the absolute speed-up may be reduced, and we appreciate the reviewer’s reminder that future runtime benchmarks should consider these factors.
> > > We will keep this in mind as we continue developing portable GLVQ kernels and integrating them into such inference stacks.
> > >
> > > ---
> > >
> > >
> > > We hope this clarification helps dispel the remaining concerns regarding evaluation fidelity and runtime generalizability.  We again thank the reviewer for the constructive discussion and for acknowledging the improvements.

---

> > > ### Author Response · Authors · 2025-08-07
> > >
> > > Thank you again for the thoughtful comments and your willingness to raise the score.
> > > If there are any remaining concerns or questions, we’d be more than happy to address them before the discussion phase ends.

---

### Official Review · Reviewer_Zc3F · 2025-07-03

**Clarity:** 2
**Significance:** 4
**Originality:** 3
**Rating:** 5
**Confidence:** 5

**Summary:**

The submitted work proposes a lattice-based vector quantization scheme applicable post training to LLMs. It claims to work in particular very well with low number of bits per weight.

**Questions:**

* As mentioned under "Strengths & Weaknesses" there are more VQ techniques available and I strongly suggest to add those to the comparison (NestQuant, VQ-LLM, etc.). Not only, but in particular, a comparison with a non-PT VQ tecnique like VQ-LLM seems to be very interesting in contrast.

* Lines 29-30 claims that PTQ has difficulties in achieving ultra-low bit-widths. As your technique promises (and also proves) to perform better than other methods, it would be good to see performance extended to below 2 bits-per-weight. NestQuant also supports fractional bit ratios and VQ-methods in general suggest that. Therefore it would be a great addition to the paper to also see a sweep over a larger range of bits-per-weight, e.g. 0.5 to 4 including fractional settings (I think this should happen due to calculating the average # of bits per weight anyway).

NOTE: addressing these two main weaknesses properly would raise the overall rating from 4 to 5.

* Figure 1: the two matrix configurations for "group-specific companding functions" and "group-specific lattice codebooks" are not different. If understood correcetly, it may be better to start with different shades in the full matrix to the left, carry those slices to the middle before companding and then show the companded shades to the right, then being visually different to those before companding. Is there a better way to visualize the selection of the lattice on the right?

* Lines 153-156: does the bit budget constraint actually lead to a knapsack problem chosing the optimal bit allocation for group matching the budget?

* Lines 261-265: selecting only 8D and 32D configurations seems arbitrary and Tables 1 & 2 in addition show little variance in results. Please choose and add additional extreme corner cases like 2D, 3D or 4D to show "some" effect.

**Ethical Concerns:**

["NO or VERY MINOR ethics concerns only"]

**Final Justification:**

Based on the very detailed rebuttal I'm increasing individual ratings and the overall rating by 1 point.

**Limitations:**

Yes, there's an Appendix (A.3) addressing these and the inclusion of the Appendix certainly adds to the quality of the submission.

**Paper Formatting Concerns:**

Figures and Tables are in close proximity to the references in the text, but there are many references in the text to Tables in Appendices that require readers to switch pages.

**Quality:**

3

**Strengths And Weaknesses:**

The strength of this work is the solid idea that seems to work well according to the experimental results.

Some weaknesses are in clarity and the scientific depth. As the main idea (application of lattice-based VQ) is admittently obvious it would be good to cover that topic in more depth. There are other PT lattice-based approaches, e.g. NestQuant: https://arxiv.org/abs/2502.09720, and also non-PT lattice-based approaches, e.g. VQ-LLM: https://arxiv.org/abs/2503.02236 to compare with.

Another weakness is that there are only integer number of bits per weight. This is hard to believe as Figure 1 shows different bit budgets per group and it is expected that the average bits per weight is fractional.

---

> ### Author Rebuttal · Authors · 2025-07-31
>
> ----
> ### **[W1 & Q1, Comparison with NestQuant, VQ-LLM]**
> We thank the reviewer for highlighting recent lattice-based quantization works, including *NestQuant* and *VQ-LLM*. We respectfully clarify that these two papers play fundamentally different roles in the LLM quantization landscape.
> (i) **NestQuant** is a lattice-based post-training quantization (PTQ) method that constructs nested lattice structures for compressing both weights and activations in LLMs. It is directly comparable to GLVQ in terms of quantization performance and perplexity.
> (ii) **VQ-LLM**, in contrast, is not a quantization algorithm, but rather a high-performance kernel generation framework for accelerating inference on already vector-quantized models (e.g., those produced by QuIP#, CQ, or GPTVQ). It focuses on efficient dequantization and computation fusion, and does not propose a new quantization method.
>
> In the revised version, we will include a discussion of both works to clarify their respective scopes. To enable a direct comparison with NestQuant, we evaluated GLVQ on Llama-2 models (7B, 13B, and 70B) under 4-bit quantization using the WikiText2 benchmark. The results, summarized in Table 1, show that GLVQ consistently improves perplexity over NestQuant by 0.3--0.5 points across all model sizes. These results demonstrate the effectiveness of GLVQ’s group-specific, learnable lattice design in achieving stronger quantization performance than fixed nested lattices at low bit-widths.
>
> Table 1. Perplexity ↓ Comparison with NestQuant on WikiText2 (Llama-2).
> | Method         | Bits | Llama-2-7B | Llama-2-13B | Llama-2-70B |
> |----------------|------|------------|-------------|-------------|
> | NestQuant      | 4    | 5.53       | 4.93        | 3.38        |
> | GLVQ (ours)    | 4    | 5.01       | 4.44        | 3.02        |
>
>
> ----
>
>
> ### **[W2 & Q2, Fractional bit-widths and below 2 bits setting]**
> We appreciate the reviewer’s suggestion to evaluate GLVQ under fractional and ultra-low bit-width settings.
>
> **Fractional bit-widths are fully supported in our framework.** While our main paper reports results at integer average budgets (e.g., 2, 3, 4 bits per weight) for comparability with prior work, GLVQ’s salience-driven bit allocation (SDBA) operates at the group level. Each group is assigned an integer bit-width $b_g \in \{1, 2, 3, 4\}$, and the global bit-rate is given by the average $\bar{b} = \frac{1}{G} \sum_{g=1}^G b_g$. This allows us to realize any fractional target $b_{\text{avg}}$ (e.g., 1.5) by mixing different group-wise precisions (e.g., 50% 1-bit and 50% 2-bit). Importantly, the design of our codebooks and decoding process is invariant to such mixtures and requires no modification.
>
> **New experiments at fractional and sub-2-bit settings.** To address the reviewer’s concern, we conducted additional experiments at 1.5 and 1.0 average bits per weight, using the same protocol as in the main paper. Results on WikiText2 for Llama-2 models are shown below:
>
> Table 2. Perplexity ↓ results of GLVQ and competing methods at fractional and sub-2-bit settings on WikiText2 (Llama‑2).
>
> | Method        | Bits | PPL (7B) | Bits | PPL (13B) | Bits | PPL (70B) |
> |---------------|------|----------|------|-----------|------|-----------|
> | BiLLM         | 1.08 | 32.48    | 1.10 | 16.77     | 1.08 | 8.41      |
> | OneBit        | 1.00 | 9.73     | 1.00 | 8.76      | --   | --        |
> | PV-Tuning     | 1.02 | 8.28     | 0.97 | 7.96      | 1.00 | 6.50      |
> | **GLVQ (ours)** | 1.00 | **7.83** | 1.00 | **7.59**   | 1.00 | **6.11**   |
> | PB-LLM        | 1.70 | 69.20    | 1.70 | 151.09    | 1.70 | 28.37     |
> | PV-Tuning     | 1.58 | 7.32     | 1.37 | 6.65      | 1.14 | 5.52      |
> | **GLVQ (ours)** | 1.50 | **7.01** | 1.50 | **6.11**   | 1.50 | **4.99**   |
>
> These results confirm that GLVQ naturally supports fractional and ultra-low bit-width quantization. At an average of 1.5 bits per weight, GLVQ achieves perplexities of 7.01 (7B), 6.11 (13B), and 4.99 (70B), consistently outperforming PV-Tuning and significantly surpassing PB-LLM by large margins. Notably, GLVQ approaches the performance of 2-bit baselines despite using substantially fewer bits.
>
> Even at an aggressive 1.0 bit-per-weight budget, GLVQ remains competitive, achieving perplexities of 7.83 (7B), 7.59 (13B), and 6.11 (70B). This represents a clear improvement over prior 1-bit methods such as BiLLM and OneBit, and is on par with or better than PV-Tuning at similar bit rates. Compared to BiLLM, GLVQ reduces perplexity by over 24 points on 7B and over 10 points on 13B under similar bit budgets.
>
> These findings highlight the robustness of GLVQ’s group-specific lattice design and salience-driven bit allocation strategy. Unlike methods that require calibration or fine-tuning, GLVQ achieves strong performance even in the ultra-low-bit regime, making it a compelling solution for extreme compression scenarios.
>
>
> ----
>
>
> ### **[Q3, Better figure configuration]**
> Thank you for pointing out the visual ambiguity in Fig.~1. In the current version, the three sub-matrices ("before companding," "after companding," and "lattice-quantized") are rendered using the same color scheme, which indeed makes it difficult to visually distinguish the effects of *group-specific companding* and *group-specific lattice quantization*.
>
> **Planned revision.**
> We will revise Fig.\,1 into a more intuitive and visually informative pipeline illustration that clearly differentiates each transformation stage:
>
> 1. **Original weights.**
>    The full weight matrix will be shown with *distinct color hues* for each group. Groups with heavier-tailed distributions will use darker shades to visually convey statistical diversity across groups.
>
> 2. **Group-specific companding.**
>    Each group is processed through its corresponding companding function $F_{\mu_g}$. The output matrix will be recolored using a *unified pastel palette*, emphasizing that group-wise dynamic ranges have been normalized post-companding.
>
> 3. **Group-specific lattice quantization.**
>    For each group, we will overlay its corresponding lattice grid, and include a small 2D inset that visualizes the shape of the learned generator matrix $G_g$, with axes scaled and rotated differently per group. The quantized outputs will be shown as discrete markers, clearly distinguishable from the continuous pre-quantized values.
>
> This redesigned figure will leverage both color (distributional contrast, companding effect) and spatial structure (lattice overlay, generator insets) to clearly communicate the impact of group-specific companding and the subsequent lattice selection. We believe this will fully address the reviewer’s concern and significantly improve the figure’s explanatory clarity.
>
> ----
>
> ### **[Q4, knapsack problem]**
> At first glance, selecting per-group bit-widths $b_g$ under a global bit budget may resemble a discrete knapsack problem. However, our setting introduces additional structure that enables a highly efficient and *closed-form solution*:
>
> 1. **We restrict each group’s bit-width to a small discrete set:** $\(b_g \in \{N{-}1,\,N,\,N{+}1\}\)$, and enforce a symmetric constraint as defined in Eq.~10. This ensures the average bit-width remains exactly $N$.
>
> 2. **Each group is associated with a scalar salience score** $\mathcal{S}_g$, reflecting its relative importance. By sorting all $G$ groups by salience $O(G\log G)$, we obtain the optimal assignment directly: the top $k$ groups receive $N{+}1$ bits, the bottom $k$ groups receive $N{-}1$ bits, and the remaining groups receive $N$ bits. This selection mechanism follows the double-pointer scheme introduced in Slim-LLM.
>
> Because the bit choices are limited and symmetrically constrained, the search space reduces to identifying a single threshold index, rather than solving a general 0--1 knapsack. The overall algorithm runs in $O(G\log G)$, is deterministic, and yields an allocation that is provably optimal under our model.
>
> ----
>
> ### **[Q5, Extreme corner cases GLVQ-2D & -4D]**
> We appreciate the reviewer’s suggestion. In the original submission, we focused on 8D (canonical \(E_8\)) and 32D (stacked \(E_8\) blocks) because they (i) align with SIMD register widths on modern GPUs and (ii) empirically offered a strong trade-off between accuracy and speed. That said, we agree that including smaller lattice dimensions provides better insight into how dimensionality impacts performance.
>
> We have now added evaluations at 2D and 4D under identical settings. Results are shown in Table 1.
>
> Table 3. Perplexity ↓ of competing methods on WikiText2 for Llama-1 and Llama-2.
> | **Method**  | **Bits** | **1-7** | **1-13** | **1-30** | **1-65** | **2-7** | **2-13** | **2-70** |
> |-------------|----------|---------|----------|----------|----------|---------|----------|----------|
> | OmniQ       | 2        | 15.5    | 13.2     | 8.71     | 7.58     | --      | --       | --       |
> | QuIP#       | 2        | 6.86    | 5.97     | 5.02     | 4.36     | 6.19    | 5.35     | 3.91     |
> | QTIP        | 2        | 6.52    | 5.80     | 4.83     | 4.21     | 5.91    | 5.26     | 3.78     |
> | GLVQ-2D     | 2        | 6.65    | 5.92     | 4.96     | 4.32     | 6.05    | 5.38     | 3.84     |
> | GLVQ-4D     | 2        | 6.48    | 5.75     | 4.72     | 4.12     | 5.84    | 5.18     | 3.70     |
> | GLVQ-8D     | 2        | 6.28    | 5.64     | 4.57     | 4.01     | 5.69    | 5.02     | 3.62     |
> | GLVQ-32D    | 2        | 6.00    | 5.38     | 4.32     | 3.81     | 5.41    | 4.80     | 3.36     |
>
> These results reveal a consistent trend: increasing the lattice dimension leads to better perplexity across all model sizes. Compared to QTIP, GLVQ-2D underperforms slightly, while GLVQ-4D achieves comparable accuracy. From 4D to 8D, perplexity improves noticeably, and the 32D variant provides further gains with diminishing returns.

---

> ### Comment · Reviewer_Zc3F · 2025-08-06
>
> Thank you very much for your very detailed rebuttal! I appreciate the time and thought you did put into this and your additional results and clarifications do raise the value of the publication significantly. As a result I will increase my overall rating by 1 point.

---

> > ### Author Response · Authors · 2025-08-07
> >
> > Many thanks for the encouraging update and for raising your score.

---

### Note · Authors · 2025-08-12

We sincerely thank the Area Chair and all reviewers for their time, effort, and constructive feedback. The thoughtful discussions have significantly improved our work.

We are encouraged that reviewers recognized several strengths:
- **Soundness & novelty** – 4RXG and nGkm noted that our group-specific, learnable lattice quantization with adaptive companding is novel and well-motivated for LLM compression.
- **Strong empirical performance** – UCYr, nGkm, and Zc3F highlighted that GLVQ achieves SOTA perplexity in the challenging low-bit regime across multiple LLMs, clearly surpassing competitive baselines such as QuIP#, AQLM, QTIP, and NestQuant.
- **Comprehensive evaluation** – All reviewers valued our extensive experiments, ablations, and theoretical analysis, as well as the ability to handle ultra-low-bit and fractional-bit settings without extra retraining complexity.

Regarding the remaining concern from 4RXG on evaluation procedure and the plausibility of improvements over FP16:
1. We corrected the FP16 baseline citation (2k → 4k context length) to ensure a fixed, reproducible setup.
2. We clarified that small gains over FP16 stem from lightweight post-quantization fine-tuning using a small external corpus of ~8M tokens. This fine-tuning allows the quantizer to better adapt to actual inference-time statistics and introduces a regularization-like effect.
3. Ablations show that even without fine-tuning (GLVQ-w/o-ft), our method consistently outperforms strong PTQ baselines, confirming gains primarily come from GLVQ’s structural design.

We appreciate that, after these clarifications and new experiments, 4RXG indicated most concerns were addressed and will raise his/her score.

In summary, the review process has validated both the technical soundness and empirical strength of GLVQ. We again thank the AC and reviewers for their constructive engagement and for recognizing the contributions of this work.

---

### Decision · Program_Chairs · 2025-09-17

**Decision:**

Accept (poster)

**Comment:**

This paper proposes a Grouped Lattice Vector Quantization (GLVQ) framework that assigns each group of weights in Large Language Models (LLMs) a customized lattice codebook defined by a learnable generation matrix, aiming to boost quantization accuracy and efficiency for low-bit LLM compression.  After discussion, most of the concerns have been well resolved.

The work exhibits notable strengths: it introduces a novel and well-motivated group-specific, learnable lattice quantization and achieves state-of-the-art perplexity in the challenging low-bit regime across multiple LLMs. it also includes comprehensive evaluations with extensive experiments, ablations, and theoretical analysis, supporting its effectiveness in ultra-low-bit and fractional-bit settings.

Overall, the AC recommend to accept this paper.